# Visualizing the interfacial-layer-based epitaxial growth process toward organic core-shell architectures

Ming-Peng Zhuo[1,2,3,5], Xiao Wei[2,5], Yuan-Yuan Li[1,3], Ying-Li Shi [1], Guang-Peng He[1], Huixue Su[2], Ke-Qin Zhang [3], Jin-Ping Guan[3], Xue-Dong Wang [1] ✉, Yuchen Wu [2] ✉ & Liang-Sheng Liao [1,4] ✉

Organic heterostructures (OHTs) with the desired geometry organization on micro/nanoscale have undergone rapid progress in nanoscience and nanotechnology. However, it is a significant challenge to elucidate the epitaxial-growth process for various OHTs composed of organic units with a lattice mismatching ratio of > 3%, which is unimaginable for inorganic heterostructures. Herein, we have demonstrated a vivid visualization of the morphology evolution of epitaxial-growth based on a doped interfacial-layer, which facilitates the comprehensive understanding of the hierarchical self-assembly of core-shell OHT with precise spatial configuration. Significantly, the barcoded OHT with periodic shells obviously illustrate the shell epitaxial-growth from tips to center parts along the seeded rods for forming the core-shell OHT. Furthermore, the diameter, length, and number of periodic shells were modulated by finely tuning the stoichiometric ratio, crystalline time, and temperature, respectively. This epitaxial-growth process could be generalized to organic systems with facile chemical/structural compatibility for forming the desired OHTs.

Organic heterostructures (OHTs) integrating intrinsic heterojunction characteristics as well as flexibility, diverse molecular structures and wide-ranging optoelectronic properties have gained impressive attention in material chemistry as promising candidates for advanced organic optoelectronics[1–3]. The carrier collection efficiency was dramatically improved by more than two orders of magnitude after combining fullerenes and poly(2-methoxy-5-(2′-ethyl-hexyloxy)−1,4-phenylenevinylene) (**MEH-PPV**) to build an internal donor-acceptor heterostructure[4]. Impressively, an expanded heterojunction area, enhanced carrier collection or desired stability/tunability of physicochemical features were achieved in such well-defined organic core-

shell heterostructures[5,6], which was beneficial for various advanced optoelectronics[7,8]. Furthermore, these optoelectronics generally demonstrate improved overall performance concerning their mono-components or physics complex[9,10]. For example, single-crystal coaxial core-shell heterostructures based on *p*-type copper phthalocyanine (**CuPc**) and *n*-type 5,10,15,20-tetra(4-pyridyl)-porphyrin (**H₂TPyP**) yielded an open-circuit voltage of ∼ 0.64 V, which is much higher than those of single-component nanowires[11].

Significantly, precise control of the morphological dimensions and structural composition of OHTs is necessary to determine their optoelectronic properties and potential applications[12,13].

[1]Institute of Functional Nano & Soft Materials (FUNSOM), Jiangsu Key Laboratory for Carbon-Based Functional Materials & Devices, Soochow University, Suzhou, Jiangsu 215123, China. [2]Technical Institute of Physics and Chemistry Chinese Academy of Sciences, Beijing 100190, China. [3]China National Textile and Apparel Council Key Laboratory for Silk Functional Materials and Technology, National Engineering Laboratory for Modern Silk, College of Textile and Clothing Engineering, Soochow University, Suzhou, Jiangsu 215123, China. [4]Macao Institute of Materials Science and Engineering, Macau University of Science and Technology, Taipa 999078 Macao SAR, China. [5]These authors contributed equally: Ming-Peng Zhuo, Xiao Wei. ✉e-mail: wangxuedong@suda.edu.cn; wuyuchen@iccas.ac.cn; lsliao@suda.edu.cn

A representative case is that the precisely prepared organic core/double-shell microwires with radial red-green-blue (RGB) emission released the miniaturized white-light emission[14]. Although various nanostructures have been widely prepared via advanced modern lithography technology, the self-assembly approach has been regarded as an indispensable alternative for the fabrication of core-shell heterostructures owing to the nonthermodynamic equilibrium state[15]. To date, a series of organic core-shell heterostructures have been constructed via purposefully selecting a material system with perfect lattice matching as well as accurate control of the dynamic assembly process[16,17]. Nevertheless, the epitaxial-growth mechanism is extremely accessible in these previous reports. In contrast, tremendous efforts are in progress to discover outstanding functions or highly performant applications[18]. It is well-surmised that the core molecules firstly self-assemble into the seeded part, then shell molecules will immediately undergo a selectively heterogeneous nucleation and epitaxial-growth process without the influence of residuary core molecules, forming the organic core-shell heterostructure (Fig. 1a). An ultralow lattice mismatch ratio below 3% is imperative for the formation of core-shell heterostructures, as in the cases of inorganic or metal species[9,15]. Nevertheless, it is still challenging to clearly illustrate the construction of organic core-shell heterostructures based on organic building blocks with a lattice mismatching ratio of more than 3%, which is rarely considered. Therefore, profoundly investigating the epitaxial-growth mechanism of organic core-shell heterostructures in the self-assembly process is of both fundamental and practical interest. It is well known that structurally similar organic molecules generally give rise to the doping process, which is purposively applied to construct uniform organic micro/nanocrystals for desired optoelectronics[19], such as white emission[20], increased carried mobility[21], and tunable energy structure[22]. In this regard, the inevitable presence of core molecules in the saturated supernatant of seed species will introduce the doping process, forming a interfacial-layer, which could act as a structural bridge and supply perfect lattice matching for facile epitaxial-growth between the core and shell parts.

Here, we propose an epitaxial-growth mechanism for the rational design and synthesis of organic core-shell heterostructures with an interfacial-layer between the core and the shell part, as presented in Fig. 1b. The presence of the interfacial-layer could effectively improve the structural/chemical compatibility and lattice matching, as well as reduce the barrier for the epitaxial-growth of the shell layer, which is conducive to the precise preparation of organic core-shell heterostructures. Interestingly, the metastable state of organic barcoded heterostructured microwires with tunable periodic shells was achieved by finely regulating the supersaturation of core and shell components, which powerfully confirms the formation of an interfacial-layer. Furthermore, elaborate modulation of the diameter, length and number of periodic shells was realized based on the controlled crystalline time, temperature, and molar ratio between core-shell species, which demonstrated the visualization of the assembly processes of organic core-shell heterostructures. This approach showed the charming universality of all organic core-shell heterostructures with the metastable state of organic barcoded heterostructure microwires prepared in the stepwise growth process, which supplies an avenue to adjust their physical/chemical properties for advanced applications.

## Results

The polycyclic aromatic hydrocarbon (PAH) of benzo[ghi]perylene (**BGP**) demonstrates a planar aromatic π-conjugated structure and a strong electron-donating ability (Fig. 2a), which is favorable for the rational design and synthesis of organic micro/nanomaterials with desired optoelectronic properties via a controlled self-assembly process[23–26]. Combined with the typical electron acceptor tetrafluoroterephthalonitrile (**TFP**) or 1,2,4,5-benzeneenetetranitrile (**TCNB**) with powerful electron affinity, **BGP** molecules could controllably self-assemble into **BGP-TFP** (**BTP**) or **BGP-TCNB** (**BTB**) cocrystals via robust charge-transfer (CT) interactions. As shown in Supplementary Figs. 1, 2, the as-prepared BTP microwires display an average diameter of 5 μm and a strong green emission with a PL peak at 515 nm, and the prepared **BTB** microwires display an average diameter of 3 μm and an intense red emission with a PL peak at 600 nm. Furthermore, the as-prepared high-crystalline **BTP** and **BTB** microwires demonstrate a growth direction along [001] and [010], respectively, which agrees with their predicted growth morphologies (Supplementary Figs. 3, 4). The π-charge-transfer from **BGP** to **TCNB** or **TFP** in these prepared **BTP** and **BTB** cocrystals was verified by Fourier Transform Infrared (FT-IR) spectra, Solid-state 13 C NMR results, electron spin resonance (ESR) spectra, and diffuse reflection absorption spectra in Supplementary Fig. 5. Notably, the **BTP** and **BTB** cocrystals demonstrated high chemical and structural compatibility (Supplementary Figs. 6, 7), which is conducive to the doping process or heterogeneous nucleation for the fine synthesis of OHTs[5,13,27]. When

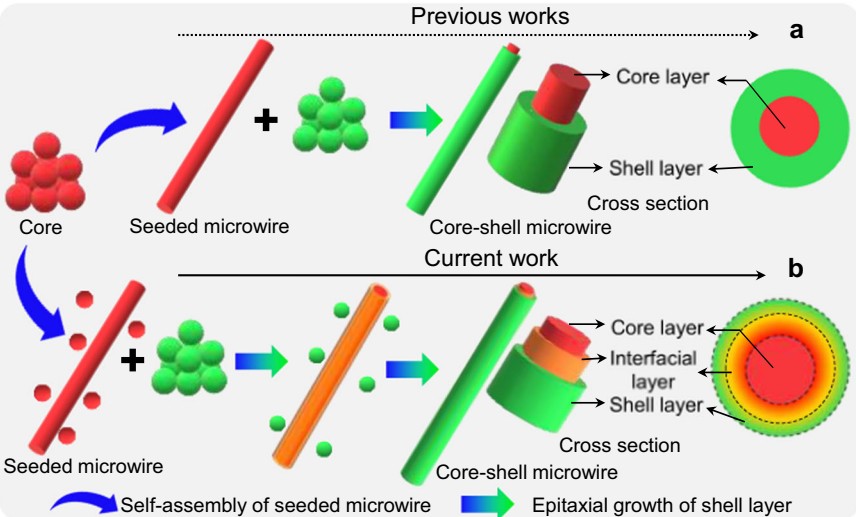

**Fig. 1 | Interfacial-layer in the epitaxial-growth for organic core-shell heterostructure.** Schematic illustrations of the preparation of the organic core-shell via (**a**) the previous perspective with the direct heterojunction between core and shell parts, as well as (**b**) the current perspective with the interfacial-layer between core and shell parts.

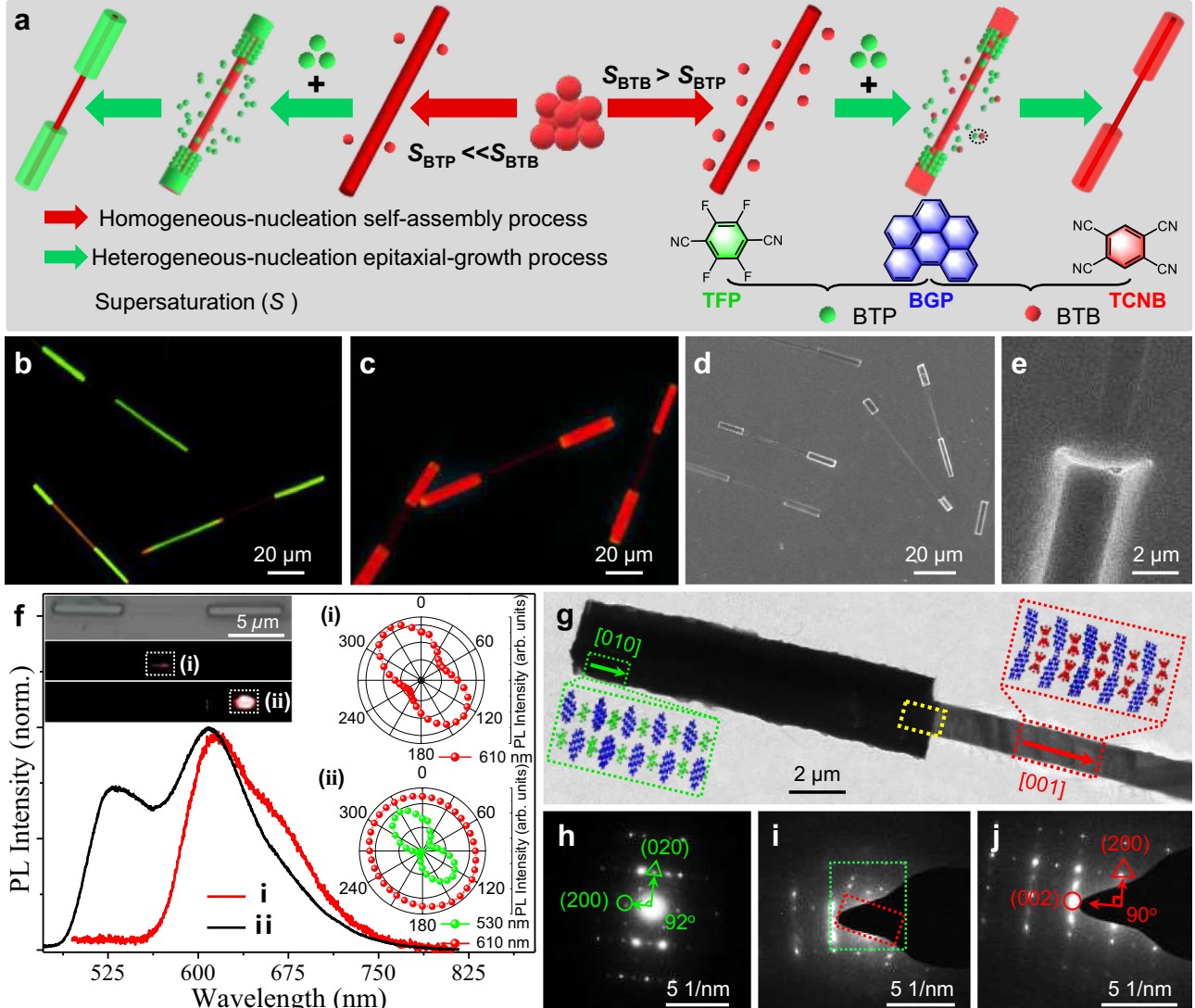

**Fig. 2 | Hierarchical self-assembly and characterizations of organic barcoded microwires. a** Schematic illustrations of the horizontal epitaxial growth process for the organic barcoded microwires. **b**, **c** FM images of the organic barcoded microwire prepared with **b** an ultrahigh and **c** a high supersaturated concentration ($S$) of BTB cocrystal. The scale bars are 20 μm. (**d**) SEM images of the organic barcoded microwire prepared with a high $S_{BTB}$. Scale bar: 20 μm. **e** SEM image of the typical tip of the organic barcoded microwire with a scale bar of 2 μm. **f** The spatially resolved PL spectra corresponding to different locations marked domains in the organic barcoded microwire. Inset: the corresponding polar image of the peak intensities. **g** TEM image of one typical organic barcoded microwire with a scale bar of 2 μm. **h**–**j** SAED patterns collected from the **h** end part (marked in green gridlines), **i** the heterojunction part (marked in yellow gridlines) and **j** the center part (marked in red gridlines) in the organic barcoded microwire of (**g**), respectively.

increasing the doping ratio of **BTB** from 0% to 10%, the prepared **BTP** microwires display a charming emission-color evolution from green to yellow and further to red (Supplementary Fig. 8), which is consistent with previous works[28,29]. It is suggested that a small quantity of **BTB** could still influence the emission color of **BTP** microwires and their hierarchical heterostructures via the doping process.

As shown in Fig. 2a, the multistep seeded growth method was employed to construct the organic barcoded microwires via self-assembly of **BTB** seed microwire and horizontal epitaxial-growth of periodic **BTP** shell layers[30]. As illustrated in Supplementary Fig. 9, supersaturation ($S$) of the **BTB** cocrystal in methanol ($5 \times 10^{-3}$ mmol/ml) is much lower than that in hexamethylene ($2.2 \times 10^{-2}$ mmol/ml), in contrast with the near supersaturation ($S$) of **BTP** in both methanol and hexamethylene ($-2.0 \times 10^{-2}$ mmol/ml). Therefore, the controlled supersaturation ($S$) of **BTP** and **BTB** was achieved via finely tuning the volume ratio between methanol and hexamethylene, determining the absence of the **BTB** doping process in the heterogeneous-nucleation epitaxial-growth of the **BTP** shell, which is consistent with the previous works

based on single-molecular building blocks[31–33]. In the solvent system with a $V_{methanol}$:$V_{hexamethylene}$ of 6:4, the higher $S_{BTB}$ than $S_{BTP}$ ($S_{BTB} > S_{BTP}$) results in a considerable **BTB** cluster in the saturated supernatant after seeded self-assembly. Then, this residue **BTB** will be doped into **BTP**, forming the red-emissive shell-layer. Upon excitation with UV light, the as-prepared organic barcoded microwires display an exposed core with deep-red emission and a shiny-red emissive shell (Fig. 2c). After further increasing the supersaturation ($S$) of **BTB** ($S_{BTB} \gg S_{BTP}$), there is not enough **BTB** in the saturated supernatant, leading to pure heterogeneous-nucleation epitaxial-growth of the **BTP** shell layer and formation of a green emissive shell-layer (Fig. 2b). As illustrated in Fig. 2c, the obtained organic barcoded microwires show exposed core microwires with deep-red emission and an orange-red emissive shell excited by green light. These organic barcoded microwires have a length of 55 ~ 70 μm with two thick shell layers at the tips and a thin core microwire at the center part, as verified by the scanning electron microscopy (SEM) image in Fig. 2d. Furthermore, the diameters of the core-shell and the exposed core microwires are 1.8 and 4.2 μm,

respectively (Fig. 2e). After $V_{\text{methanol}}$:$V_{\text{hexamethylene}}$ transitions to 7:3 and 8:2, the shell layers of the organic barcoded microwires display orange and yellow emission excited by UV light (Supplementary Fig. 10). Moreover, these organic barcoded microwires only present red-emissive core microwires upon excitation with green light, and the corresponding nonemissive shell layers are due to the low quantity of **BTB** in the **BTP** shell layer. The spatially resolved spectrum collected at location (i) at the exposed core part of the organic barcoded microwires by focusing a laser beam ($\lambda = 375$ nm) shows red emission with a PL peak at 600 nm (Fig. 2f), which is consistent with that of the **BTB** cocrystal (Supplementary Fig. 2). Additionally, the additional green light with a PL peak at 515 nm emitted from the **BTP** shell emerges on the PL spectrum of location (ii) at the core-shell part. With a low doping ratio of **BTB** in the shell layer, obvious green light is found in the corresponding PL spectrum of the shell layer (Supplementary Fig. 11), suggesting that the red emission of the shell layer comes from the doped **BTB**. Notably, the polarized angles $\theta$ of red light (600 nm) emitted from the exposed core and green light (525 nm) emitted from core-shell parts are both 30° (inset of Fig. 2e), which powerfully confirms the analogous crystal structures of core and shell parts in the organic barcoded microwires[6]. The corresponding highest PL intensities of the red light (600 nm) emitted from the core-shell at the end part hardly change with the polarization angle (inset of Fig. 2f). As we well know the organic microcrystals generally display the PL anisotropic, which is attributed to the anisotropic molecular packing in the crystal structure. It is indicated that the red emission of the core-shell part contributes to the doping of the red-emissive **BTB**. Impressively, these prepared organic barcoded microwires still remain the barcoded structure and designated emission feature after stored in air for 90 days (Supplementary Fig. 12), which powerfully demonstrates their excellent stability. As illustrated in the transmission electron microscopy (TEM) image of Fig. 2g, the shell layer partly coats the end part of the core microwires with smooth surface feature, which is consistent with the representative SEM image (Fig. 2e). The selected area electron diffraction (SAED) pattern of the shell layer at the end part (green gridline) presents an identical pattern as that of the **BTP** microwire (Fig. 2h and Supplementary Fig. 3b). This indicates epitaxial-growth of the **BTP** shell layer on the surface of the core microwires along the [011]. In contrast, the SAED pattern of the exposed core (red gridline) agrees well with that of the **BTB** microwire (Fig. 2j and Supplementary Fig. 3d), suggesting a core growth direction of [010]. Notably, the SAED pattern of the heterojunction (yellow gridline) exhibits identical patterns to those of both **BTP** and **BTB** cocrystals (Fig. 2i), confirming the simultaneous presence of **BTB** and **BTP** cocrystals. The **BTB** and **BTP** cocrystals hold analogous molecular parking along the growth direction driven by the CT interaction. Combined with the low lattice mismatch ratio ($f$) of 0.8% ($d_{\text{BTP}}^{(100)} = 9.15$ Å $\approx d_{\text{BTP}}^{(100)} = 9.22$ Å, Supplementary Table 1), their high chemical and structural compatibility facilies facet-selective heteronucleation and epitaxial-growth for the construction of OHTs[13,27]. These results indicate that the **BTB** was firstly self-assembled into seeded microwires acting as a core part via CT interaction along the [010] direction. Then, the **BTP** horizontally epitaxial grows on the end surface of the seeded core, forming the organic barcoded microwires.

Due to the rapidly increased solution temperature and solvent volatilization caused by light sources during the fluorescence microscopic imaging process, it is still a challenge to recode the in situ visualization of hierarchical self-assembly of the organic barcode microwires. To further investigate the hierarchical self-assembly mechanism of the organic barcoded microwires, the time-based statistical fluorescence microscopy (FM) and SEM techniques were employed to accurately quantify the time-based epitaxial-growth of the **BTB**-doped **BTP** shell layer with red emission. The storage time ($t$) for the horizontally epitaxial-growth of the **BTP** on the surface of the **BTB** seeded microwires is crucial for the fine synthesis of the organic barcoded microwires[34]. After adding **BTP** stock solution into the

growth solution system ($V_{\text{cyclohexane}}/V_{\text{Methanol}} = 6:4$) of the **BTB** microwires stored for 2.5 min, shiny-red emissive shell layers appear at the tip surface of the deep-red emissive **BTB** microwires, as evidenced by the FM and SEM images in Supplementary Fig. 13. With a storage time of 10 min, shiny-red emissive shell layers with small thickness were observed on the end part of the deep-red emissive **BTB** microwires (Fig. 3a, d). As shown in the high-resolution SEM image of Fig. 3g, the diameters of the exposed core and the complete core-shell parts are 2.1 and 2.9 μm, respectively. Further increasing the storage time to 15 min, the length and thickness of the shiny-red emissive shell layers display a considerable enhancement (Fig. 3b, e). The exposed core retains the initial morphology with a diameter of 2.2 μm, while the diameter of the complete core-shell parts increases to 4.8 μm (Fig. 3h). With a storage time of 20 min, the shell thickness with strong shiny-red emission increases to 8.1 μm, in contrast to the exposed core with a diameter of 2.0 μm (Fig. 3c, f, i). Furthermore, the exposed core microwires display an unaltered diameter of ~2.0 μm during the whole epitaxial-growth process (Fig. 3j), which is agreed with the representative units as shown in Fig. 3g–i. Meanwhile, the diameter of the complete core-shell parts at the end of the organic barcoded microwires demonstrates an increase from 2.0 to 8.1 μm during increasing the storage time from 0 to 20 min. With a storage time of more than 20 min, the diameter of the complete core-shell parts also remains unchanged. However, the diameter of the exposed core part is still ~2 μm, showing a nondependence on the storage time. It is suggested that the horizontally epitaxial-growth of the BTP on the surface of the seeded **BTB** microwires is finished with a storage time of more than 20 min. Therefore, the precise preparation of organic barcoded microwires with a tunable shell layer at the end part is rationally controlled by adjusting the storage time of **BTP** epitaxial-growth.

Significantly, the exposed degree of the core microwire at the center part in the organic barcoded microwires could be rationally adjusted by modulating the molar ratio between **BTP** and **BTB** ($\eta = n_{\text{BTB}}/n_{\text{BTP}}$) in the horizontally epitaxial-growth process (Fig. 4a). As illustrated in Fig. 4b, shiny-red emissive shell layers with a short length of ~10 μm emerge on the end part of the deep-red emissive core microwires with a molar ratio $\eta$ of 8:3. After changing the excitation from UV light to green light, the shiny-red emissive shell layers are replaced by bright red tips (Supplementary Fig. 14a), which is attributed to the formation of the interfacial-layer via **BTB** doping in the epitaxial-growth process of the **BTP** shell layer. Furthermore, a brick-like microrod with a length of ~8 μm and a width of ~4 μm was found on the two tips of core microwires with a length of ~80 μm (Supplementary Fig. 15a, b). After increasing the molar ratio $\eta$ from 8:3 to 8:4, the shiny-red emissive shell layers demonstrate a length of ~40 μm, illustrating a decreased degree of exposure of the core microwires at the center part (Fig. 4c, Supplementary Fig. 14b, c). There is an obvious reduction in the exposed degree of the core microwires at the center part with a molar ratio $\eta$ of 8:5 (Fig. 4d, Supplementary Figs. 14c, 15d). The exposed core microwires are only observed at the center part with a molar ratio $\eta$ of 8:6 (Fig. 4e, Supplementary Figs. 14d, 15e). Impressively, the exposed core microwires disappeared in these prepared organic barcoded microwires with a molar ratio $\eta$ of 8:7 and were replaced by shiny-red emissive microwires with a distinct heterojunction between the two shell layers at the center part upon excitation with UV light (Fig. 4f and Supplementary Fig. 16a$_1$). After transforming the excitation from UV light to green light, strongly red-emissive core microwires coated by a weak red-emissive shell layer with a joint-like heterojunction were found (Figs. 4g, i, and Supplementary Fig. 16a$_2$). The SEM images (Fig. 4h) clearly illustrate that these prepared OHT microwires exhibit a smooth surface and a clear heterojunction. The anisotropic epitaxial-growth of the **BTP** shell layer at the two tips generally results in a small difference in the diameters of the two shells at the heterojunction (Fig. 4j,

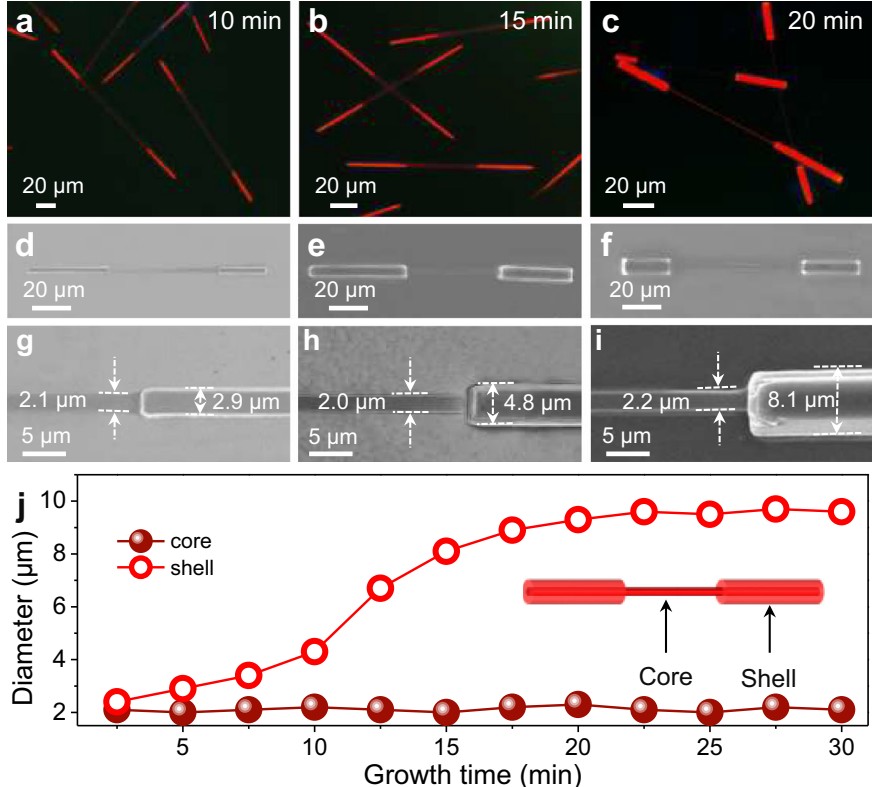

**Fig. 3 | Elaborate control of organic barcoded heterostructure. a–c** FM and **d–i** SEM images of organic barcoded microwires prepared with epitaxial growth times of (**a**, **d**, and **g**) 10 min, (**b**, **e**, and **h**) 15 min, and (**c**, **f**, and **i**) 20 min. The scale bars of (**a–f**) and (**h**, **i**) are 20 and 5 μm, respectively. **j** Plot of the diameter of core and shell parts versus the epitaxial growth time.

Supplementary Fig. 16b, c). All of these results powerfully confirm the formation of a representative core-shell structure via sufficient horizontal epitaxial-growth of the **BTP** shell layer from the tip to the center parts on the surface of the **BTB** microwires, as illustrated in Fig. 4a[17]. Furthermore, the diameter of the exposed wires in these prepared organic barcoded microwires could be controled by finely adjusting the concentration of the **BTB** and **BTP** stock solution in the hierarchical self-assembly as elucidated in Supplementary Fig. 17.

Notably, the prepared organic core-shell microwires with a tunable emission color from red to green could also be controlled after epitaxial-growth of pure **BTP** by finely increasing the molar ratio $\eta$ from 8:7 to 8:10 (Fig. 4a). With a molar ratio $\eta$ of 8:8, the prepared organic core-shell microwires with a smooth surface display a uniform shiny-red emission without the obvious heterojunction excited by UV light (Fig. 4k, Supplementary Fig. 18a, c), confirming the formation of the complete shell layer. Upon excitation with green light, the prepared organic core-shell microwires exhibit a thin core microwire with a red-emission coating by a weak red-emission microrod (Fig. 4l and Supplementary Fig. 18b). Owing to the high defect density at the heterojunction inducing the increased optical scattering, these prepared organic core-shell microwires show a strong red emission at the center part. It is powerful to affirm that the horizontal epitaxial-growth of the shell layer from the end tip to the center part induces the formation of organic core-shell microwires. With a molar ratio $\eta$ of 8:9, the prepared organic microwires with a regular rod-like morphology were successfully obtained (Supplementary Fig. 18c). The prepared organic microwires show an obvious green-emissive shell layer with a small thickness on uniform yellow-emissive microwires excited by UV light (Supplementary Fig. 19a). Under green-light excitation, a strong red-emission core and weak red-emission shell are found in these prepared microwires, suggesting a characteristic core-shell

structure (Supplementary Fig. 19b). The spatially resolved PL spectrum of the as-prepared OHT microwire (Supplementary Fig. 20a) involves green light ($\lambda = 515$ nm) from **BTP** and red light from BTB ($\lambda = 600$ nm). Furthermore, the highest PL intensities of the green emission from the **BTP** shell layer and the red emission from the **BTB** core part are polarization dependent on the rod axis with the same polarized angles, which confirms the high crystallinity of the shell and core parts. Combined with the corresponding XRD pattern with the characteristic diffraction peaks of both **BTP** and **BTB** (Supplementary Fig. 19d), these powerful results indicate the successful synthesis of the organic core-shell structure. Furthermore, the yellow emission of the organic core-shell microwires evolves into bright green emission (Fig. 4m) after horizontal epitaxial-growth of the pure **BTP** shell layer with a molar ratio $\eta$ of 8:10. With the excitation of green light, a well-defined core-shell structure with a typical trace of the heterojunction was found, as evidenced by the FM image in Fig. 4n. Analogously, organic coreshell microwires without the typical trace of the heterojunction could also be prepared with a molar ratio $\eta$ of 8:10 by applying pure hexamethylene as a poor solvent (Supplementary Fig. 21). Therefore, the elaborate control of molar ratio $\eta$ is crucial to rationally modulate the exposed degree of the core at the center part of the organic barcoded microwires as well as the emission color of the organic core-shell microwires.

It is well known that temperature is an important factor in the self-assembly process to precisely regulate the corresponding thermodynamic and kinetic growth pathways, forming the desired morphology and heterostructures[6,35,36]. The controlled number and the heteronucleation location of the periodic shell layer of organic barcoded microwires could be achieved by tuning the temperature for controllably selective heteronucleation and horizontal epitaxialgrowth. As illustrated by the FM and SEM images in Fig. 5a, the

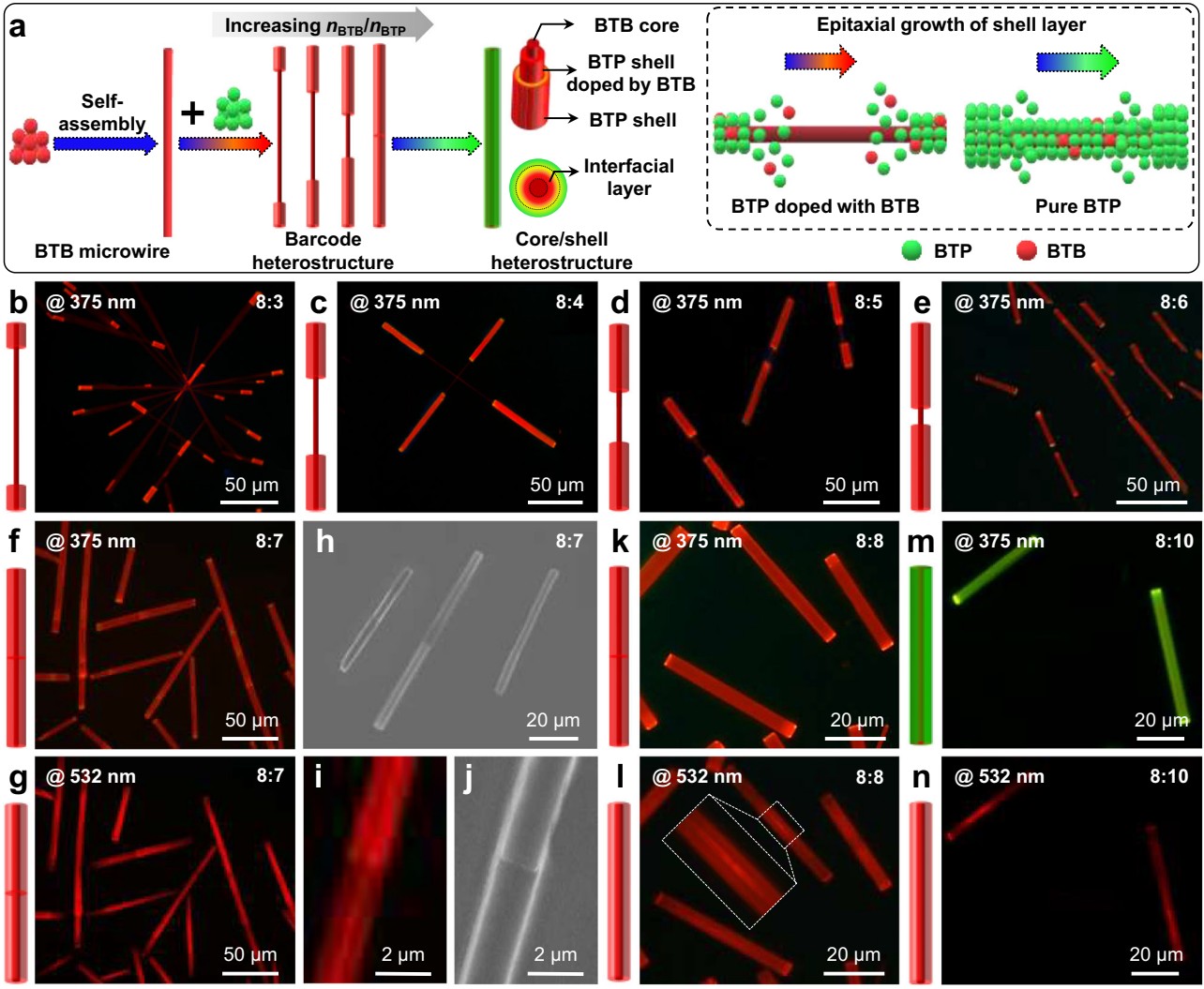

**Fig. 4 | Morphology evolution of epitaxial-growth for forming organic core-shell heterostructure. a** Schematic illustrations of the morphology evolution from barcoded to core-shell heterostructures via increasing the molar ratios between BTB and BTP cocrystals. **b**–**e** FM images of organic barcoded microwires excited with the UV band based on the different molar ratios between BTB and BTP: **b** 8:3, **c** 8:4, **d** 8:5 and **e** 8:6, scale bars are all 50 μm. **f**, **g** FM and **h** SEM images of organic core-shell microwires with an obvious heterojunction between the two shell layers at the center part. The scale bars of (**f**, **g**) and (**h**) are 50 and 20 μm, respectively. **i** FM and **j** SEM images of the typical heterojunction between the two shell layers at the center part with a scale bar of 2 μm. **k**–**n** FM images of organic core-shell microwires based on different molar ratios between BTB and BTP: **k, l** 8:8 and (**m, n**) 8:9. (**f, k,** and **m**) and (**g, i, l,** and **n**) were excited with the UV and green bands, respectively.

organic barcoded microwires with the red-emissive shell layer at the two end parts of a slender core microwire were prepared with the stock solution at a temperature of 30 °C. Remarkably, a well-defined core-shell structure was recognized at the two ends under green-light excitation (Fig. 5a(ii)). The organic barcoded microwires with an additional shell layer at the center parts were prepared with the stock solution at a temperature of 35 °C (Fig. 5b). In contrast, the enhanced temperature of the stocking solution leads to an enhancement in the barcoded number from 2 to 3, as well as an obvious decrease in the corresponding length and diameter of the periodic shell layer. The polarization images of the PL emission from the center part coated by the shell layer and the exposed core part further confirm the construction of the crystalline organic barcoded microwires with the periodic shell (Supplementary Fig. 22). Moreover, organic barcoded microwires with four barcoded structures were obtained based on the stock solution at 40 °C (Fig. 5c). As shown in the SEM image in (iv) of Fig. 5c, the diameter of the periodic shell decreases when enhancing the corresponding number. After increasing the temperature of the stock

solution to 45 °C and 50 °C, the unidentifiable obtained organic barcoded microwires with additional green-emissive **BTP** microcrystals were obtained, which demonstrated an increase in the number of the barcoded blocks (Supplementary Fig. 23).

The evolution of organic barcoded microwires with a tunable number and diameter of the periodic shell layer was achieved by rationally adjusting the temperature of the stock solution in the selective heteronucleation process. After nucleation and growth at the early stage, **BTB** self-assembled into seeded microwires with high defect density at tips owing to the rapid self-assembly. The BTP supersaturated solution was applied as the shell solution and added into the self-assembly system of the **BTB** seeded microwires. Then, the **BTP** will undergo selective heteronucleation and horizontal epitaxial-growth on the surface of the **BTB** seeded microwires, forming the organic barcoded microwires (Fig. 5d). Importantly, the surface free energy $E_{(hlk)s}^{surf}$ of the exposed crystal planes in the growth morphology of **BTB** microwires follow the order: $|E_{(011)s}^{surf}| = |E_{(0-11)s}^{surf}| (-111.71 \, \text{kcal/mol})| > |E_{(110)s}^{surf}| = |E_{(0-10)s}^{surf}| (-79.82 \, \text{kcal/mol})| > |E_{(020)s}^{surf}| (-34.29 \, \text{kcal/mol})|$. In order to reduce the highest surface free

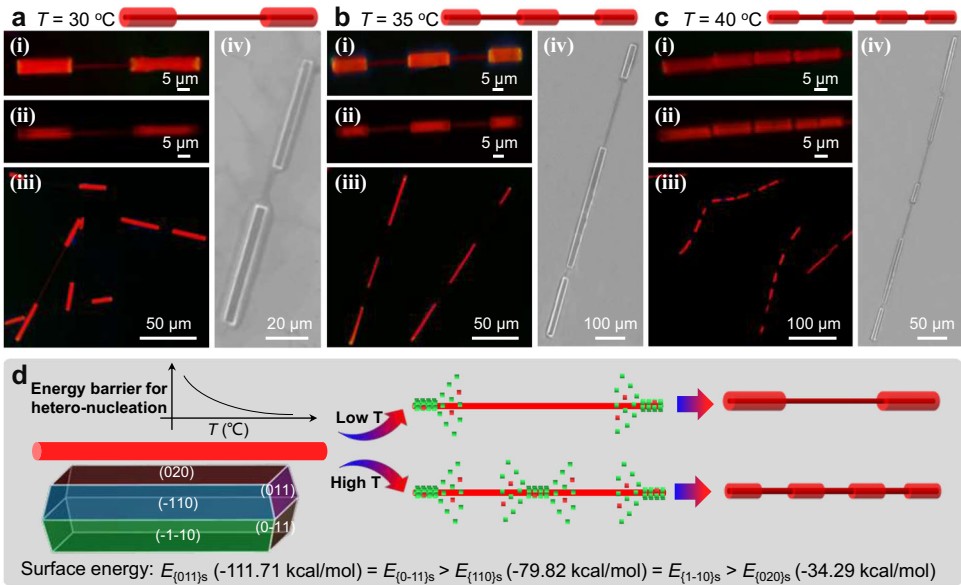

**Fig. 5 | Growth mechanism of the organic barcoded heterostructure.**
**a**–**c** Morphology images of organic barcoded microwires based on the temperature of epitaxial growth process: (**a**) 30 °C, (**b**) 35 °C, and (**c**) 40 °C. (i-iii) are FM images. The scale bars of (i and ii) in (**a**–**c**) are all 5 μm. The scale bars of (iii) in (**a**–**c**) are 50 and 100 μm, respectively. (iv) SEM image. The scale bars of (**a**–**c**) are 20, 100, and 50 μm, respectively. **d** Schematic illustrations of the horizontal epitaxial growth process for the formation of organic barcoded microwires with different barcoded numbers.

energy for surface-interface energy balance, **BTB** is favorable to selectively heteronucleate on the crystal planes of (011)s and (0−11)s at the tips (Supplementary Fig. 24a, b). Subsequently, the horizontal epitaxial-growth of **BTP** on the side surface of the crystal planes of (110)s and (1−10)s is carried out, as verified by the SEM image in Supplementary Fig. 24d. Finally, the horizontal epitaxial-growth of **BTP** on the top surface of the crystal planes of (020)s (Supplementary Fig. 24c) facilitates the formation of the whole shell layer at low temperatures. The high temperature decreases the energy barrier for heteronucleation, which has been successfully used for controllable Plateau–Rayleigh crystal growth[29]. Accordingly, the high temperature could effectively decrease the energy barrier to promote the heteronucleation of the **BTP** on the center surface of the **BTB** core microwires, forming organic barcoded microwires with several periodic shells. Furthermore, the increased temperature could induce a high evaporation rate of the organic solvent, resulting into the high supersaturation concentration of **BTP** cocrystals for the rapid self-assembly process. During this process, there are not enough time the subsequently formed **BTP** nucleus to choose a topological configuration with minimized overall surface and interfacial energy, inducing the epitaxial-growth of abundant **BTP** cocrystal containing **BTB** on the side surface of **BTB** seeded wires with a larger surface to volume ratio, showing an increased shell number.

The highly ordered organic single-crystal arrays demonstrated the fascinating on-demand photons/electrons transport features, which presents an enticing prospect for the large-scale and integrated optoelectronic application[37]. Organic barcoded microwire arrays were successfully prepared via a capillary-bridge confined assembly method as illustrated in Fig. 6a. Typically, the **BTB** stock solution was dropped onto the substrate and then covered by an asymmetric-wettability micropillar template (Supplementary Fig. 25), resulting in a sandwich-type structure (Supplementary Fig. 26) as the cases in the previous works[38]. During the solvents gradually evaporating, the liquid film ruptures and forms the capillary bridges attached to micropillars (Supplementary Fig. 26), which display the blue emission undergoing the (I) supersaturation and (II) nucleation processes in the kinetics of **BTB** self-assembly process as presented in Fig. 6g and Supplementary Fig. 27a. Driving

by the strong CT interaction, the supersaturated **BTB** could self-assembled into the red-emissive **BTB** microwire arrays corresponding the (III) growth process in the kinetics of **BTB** self-assembly process (Fig. 6e and Supplementary Fig. 27b) at the tri-phase contact line after the dewetting of capillary bridges. In order to obtained the desired barcoded structure arrays, the **BTP** stock solution was added into the above system before the solvent thoroughly evaporated. Following the **BTP** crystal heteronucleation and growth process with the **BTB** doped process on the surface of the **BTB** microwire array, the organic barcoded microwire arrays with the red-emissive shell layer on the surface of a slender core microwire were successfully prepared as verified in Supplementary Fig. 27d. These prepared organic barcoded microwires present the typical barcoded heterostructures similar to that obtained by the solution self-assemble method. Due to being at (III) state, the additional **BTP** stock solution could not affect the morphology of the **BTB** microwire arrays. Furthermore, the residuary **BTB** component firstly will dope the **BTP** to form the interfacial-layer.

Notably, the temperature also has a crucial effect for the controlled fabrication of the organic barcoded microwire arrays, expressly, the distance between the adjacent shell layer (named barcoded distance) as verified in Fig. 6b–d. As illustrated by the FM images in Fig. 6b, the organic barcoded microwire arrays were prepared with the system temperature of 30 °C. The barcoded distance mostly concentrates at the range 52 - 54 um as confirmed in Fig. 6e. As shown in the FM images (Fig. 6c) with same resolution compared with Fig. 6b, there are three shell layers with the strong red-emission appeared on the slender core microwire after increased the system temperature to 35 °C, whose barcoded diameter mostly concentrates at the range of 19 - 21 um (Fig. 6e). The organic barcoded microwire arrays with the dense and red-emission shell layers on the surface of a slender core microwire were controllably prepared via further increasing the system temperature to 40 °C (Fig. 6d). The corresponding barcoded diameter at the range of 3 - 4 um (Fig. 6e), which are shorter than those of these prepared organic barcoded microwires with the system temperature at 30 and 35 °C. Impressively, the concentration of **BTP** stock solution also plays a crucial role for controlled preparation of organic barcoded microwire arrays (Supplementary Fig. 28). The

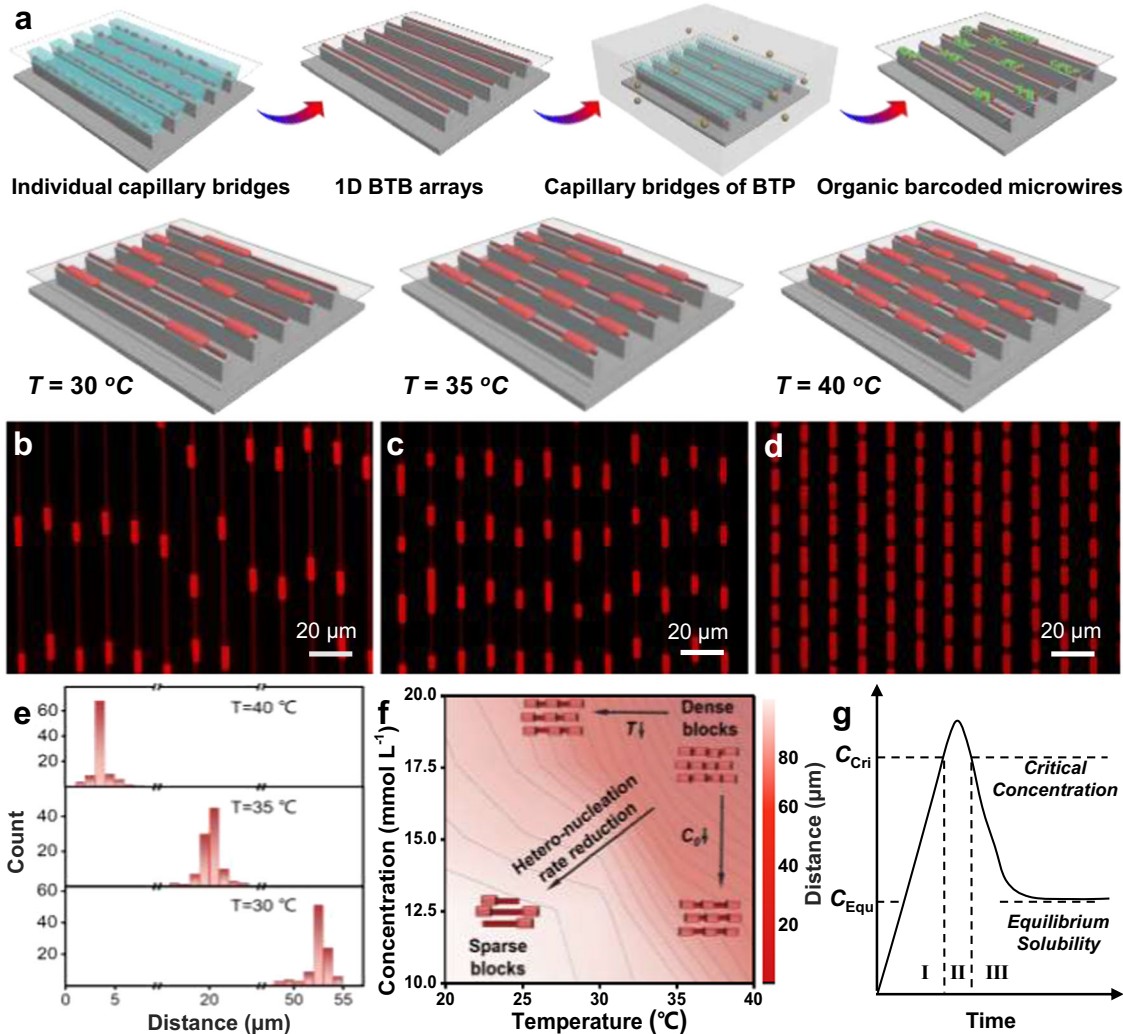

**Fig. 6 | Fine synthesis of highly ordered organic barcoded microwires.**
**a** Schematic representation (top) and fluorescence observation of the dewetting
process of capillary-bridge assembly system. **b**, **c** Morphology images of organic
barcoded microwires arrays based on the temperature of a capillary-bridge con-
fined assembly method: **b** 30 °C, **c** 35 °C, and **d** 40 °C. The scale bars are 20 μm.
**e** The distribution of the core microwire distance between the two shell layer.
**f** Contour plots of the core microwire distance between the shell layer assembled
by different concentration of BTP stock solution at different temperatures.
**g** Kinetics of solute supersaturation including (I) supersaturation, (II) nucleation,
and (III) growth. Precursor conversion reactions that limit the crystallization
determine the temporal evolution of monomer concentration as well as the steady
state supersaturation during the growth phase.

contour map of barcoded distance indicates the synergistic effect of
temperature and concentration in second self-assembly process on the
regulation of density and location of the periodic shell layer (Fig. 6f).
The barcoded distance from 3 to 50 um could be precisely controlled
by tuning the temperature and concentration of **BTP** stock solution.
The enhanced temperature and increased concentration lead to more
heteronucleation of the **BTP** on the center surface of the **BTB** core
microwires due to the reduced energy barrier, forming organic bar-
coded microwires with dense periodic shells. Significantly, the low **BTP**
concentration will dissolve the self-assembled **BTB** microwire arrays
and could not obtain the desired barcoded microwire arrays (Sup-
plementary Figs. 27e–h, 28). In conclusion, multiple levels of confined
assembly method could fabricate highly ordered organic barcoded
microwires. These organic barcoded microwire arrays provide great
advantages and promising in the field of precisely regulating photon
transmission for organic photonic circuits.

Impressively, this strategy for the organic barcoded hetero-
structures could be generalized to all organic systems, clarifying an
attractive universality. As shown in Fig. 7a, the electronic acceptors of
octafluoronaphthalene (**OFN**) and 7,7,8,8-Tetracyanoquinodimethane
(**TCNQ**) demonstrate different electron affinities compared with **TClP**

and **TCNB**[39]. Compared with the electronic donor **BGP**, benzo[c]phe-
nanthrene (**BCP**) exhibits a typical π-conjugate structure and strong
electron-donating ability. Depending on these introduced π-
conjugated molecules, desired organic microwires with tunable
emission colors from blue to red and further to near-infrared (NIR)
were successfully prepared, as shown in Fig. 7b. Typically, **BGP** and
**OFN** could self-assemble into cyan-emissive **BGP-OFN** (**BON**) micro-
wires with a CIE of (0.18, 0.43) and a high crystallinity, as verified in
Fig. 7b and Supplementary Fig. 29. Similarly, blue-emissive **BCP-TClP**
(**BPP**) microwires with a CIE of (0.20, 0.49) (Supplementary Fig. 30)
and NIR-emissive **BCP-TCNQ** (**BTQ**) with a CIE of (0.73, 0.27) (Sup-
plementary Fig. 31) were successfully prepared via a solution self-
assembly process. Furthermore, the supersaturated solution of **BON**
cocrystals as the shell solution was added into the self-assembly sys-
tem of **BTB** microwire seeded microwires, and then the mixed solution
was quickly dropped on a quartz plate. The **BTB-BON** organic bar-
coded microwires were observed after the solvent evaporated com-
pletely. Upon excitation with UV light, the **BTB-BON** barcoded
heterostructure with red-emissive core microwires and two cyan-
emissive shell layers at the end tips was clearly observed, as shown in
Fig. 7d (i). As given in Fig. 7d (ii), the spatially resolved PL spectra

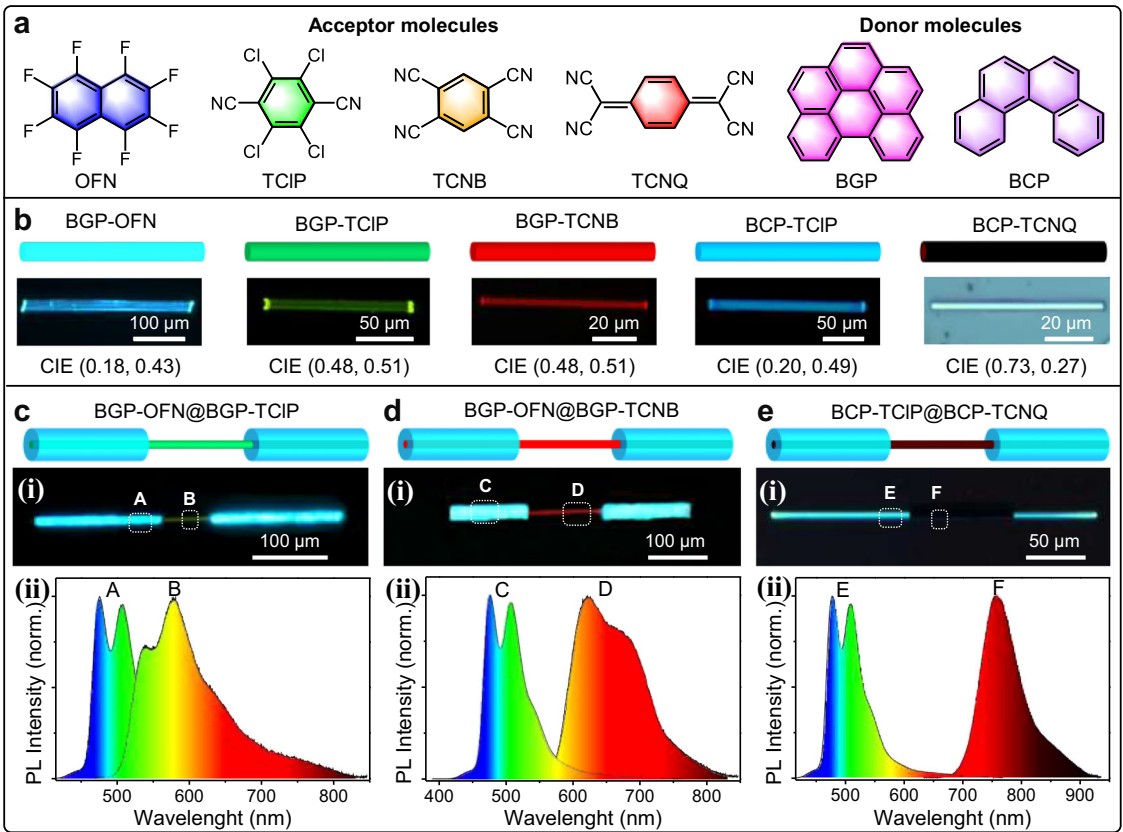

**Fig. 7 | Universality of interfacial-layer in the epitaxial-growth for organic barcoded heterostructures. a** Schematic showing the molecular structures of four electronic acceptors of OFN, TClP, TCNB, and TCNQ, as well as two electronic donors of BGP and BCP. **b** Morphology images of organic CT cocrystals of BGP-OFN, BGP-TClP, BGP-TCNB, BCP-TClP, and BCP-TCNQ microwires. The excitation of the corresponding FM images is UV light. **c, d** Organic barcoded microwires based on different combinations of organic CT cocrystals: **c** BGP-OFN@BGP-TClP, **d** BGP-OFN@BGP-TCNB, and **e** BCP-TClP@BCP-TCNQ. (i) FM image of the organic bar-coded microwire, and (ii) the spatially resolved PL spectra corresponding to different locations marked domains in (i).

collected from the periodic shell layer (marked as C) and exposed core part (marked as D) in Fig. 7d (i) are consistent with those of the cyan-emissive **BON** microwire and red-emissive **BTB** microwire, respectively (Supplementary Fig. 2). This illustrates that the selectively heterogeneous nucleation and horizontally epitaxial-growth of **BON** cocrystals on the surface of the prepared **BTB** microwires acting as seed parts are favorable for the formation of organic barcoded microwires. In contrast to the red-emissive **BTB** microwires, the green-emissive **BPP** microwires could also be applied as a core part for the horizontally epitaxial-growth of cyan-emissive **BON** cocrystals, forming a **BPP-BON** barcoded structure, as illustrated in Fig. 7c. Similarly, organic barcoded microwires with periodic shell layers of cyan-emissive **BON** and the core part of deep-red emissive **BTQ** or green-emissive **BPP** cocrystals (Fig. 7e and Supplementary Fig. 32) were successfully prepared. Thus, the periodic shell layers epitaxially grown on the seeded surface based on an interfacial-layer at the heterojunctions have been demonstrated to be a universal process to achieve barcoded and core-shell heterostructures, as in the case of other kinds barcoded heterostructures based on organic halogen cocrystal of **DPEpe-F₄DIB** and organic hydrogen cocrystal of **DPEpe-BrFTA**[6]. Furthermore, the formation of interfacial-layer in the epitaxial-growth present an attractive universality in the sequential crystallization process (Supplementary Fig. 33), the physical vapor deposition (PVD) for forming OHTs (Supplementary Fig. 34).

## Discussion

In summary, we have demonstrated a growth phenomenon that is fundamentally importance to organic core-shell heterostructures prepared via an epitaxial-growth process in hierarchical self-assembly:

the presence of an interfacial-layer between core and shell layers. The fascinating flexibility and diversity of organic semiconductors facilitate the doping process to form the interfacial-layer, which could effectively enhance structural/chemical compatibility to achieve perfect lattice matching for the facile epitaxial-growth of organic core-shell heterostructures. In the stepwise growth process, organic barcoded microwires with periodic shells in a metastable state were successfully obtained by finely adjusting the supersaturation of **BTP** and **BTB** depending on the controlled volume ratio between methanol and hexamethylene. Furthermore, the longitudinal epitaxial-growth of the periodic shell along the rod-axil to center part results in the morphology evolution from barcoded to core-shell heterostructures, showing a distinctive assembly visualization of organic core-shell heterostructures. Notably, the diameter, length, and nucleation location of the periodic shell are proportional to the crystalline time, the molar ratio between core-shell species, and the temperature. Such an assembly visualization with an interfacial-layer and a metastable state of organic barcoded microwires also exists in all organic systems with high structural/chemical compatibility, which provides promising insights into epitaxial-growth for forming OHTs.

## Methods

### Controllably self-assembly of organic BGP-based CT cocrystal microwires

In a typical experiment, 20 ml **BTP** stock solution with a concentration of 10.0 mmol L⁻¹ (containing 0.2 mmol (55.26 mg) **BGP** and 0.2 mmol (40.00 mg) **TFP** in DCM at room temperature) was added to 20 mL ethanol, and then dropped onto a quartz substrate. After the solvents completely evaporated, the **BTP** microwires were

observed. Likewise, the 220 mL **BTB** stock solution with the same concentration of 10.0 mmol L$^{-1}$ (containing 0.2 mmol (55.26 mg) **BGP** and 0.2 mmol (35.62 mg) **TCNB** in DCM at room temperature) was added into 20 mL ethanol. Then, the mixture was directly dropped onto a quartz substrate, and the **BTP** microwires via a self-assembly process.

**Hierarchical self-assembly of the organic barcoded microwires**
20 ml cyclohexane was added in 80 ml methanol to form the mixed solution acting as a poor solvent. The 1 ml **BTB** stock solution with a concentration of 20 mmol/L at room temperature was added into the 10 mL mixed solution, and stored for 1.5 h at room temperature. Then, 1 ml **BTP** stock solution with a concentration of 20 mmol/L was added into the above stored mixed solution and quickly added into the above saturated supernatant on quartz substrate. Finally, the typical organic barcoded microwires were successfully prepared. Furthermore, the diameter, length, and number of periodic shells were modulated by finely tuning the stoichiometric ratio from 8:3 to 8:9, the crystalline time from 10 to 20 min, and the temperature from 30 to 40 °C, respectively.

## Data availability
The authors declare that all relevant data supporting the findings of this study are available in the paper and its Supplementary Information files or from the corresponding authors upon request. Source data are provided with this paper.

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

## Acknowledgements
The authors acknowledge the financial support from the National Natural Science Foundation of China (nos. 52173177 (X.-D. W.), 21971185 (X.-D. W.) and 52203234, (M.-P. Z.)), the Natural Science Foundation of Jiangsu Province (no. BK20230010, BK20221362 (X.-D. W.)), "Textile Vision Basic Research Program" (J202310, M.-P. Z.), and the Science and Technology Support Program of Jiangsu Province (no. TJ-2022-002 (X.-D. W.)). Furthermore, this work is supported by the Suzhou Key Laboratory of Functional Nano & Soft Materials, Collaborative Innovation Center of Suzhou Nano Science & Technology, Key Laboratory of Flame Retardancy Finishing of Textile Materials (CNTAC), the 111 Project, Joint International Research Laboratory of Carbon-Based Functional Materials and Devices, and Soochow University Tang Scholar.

## Author contributions
X.-D.W., Y.W., and L.-S.L. conceptualized the experiments; M.-P.Zh., H.S., Y.-Y.Li., G.-P.H. and X.W., synthesised the organic micro/nanostructures and determined the structural/optical characteristics. M.-P.Z. and Y.-L.S. performed the simulation of the predicted morphologies of organic molecules. M.-P.Z., K.-Q.Z., J.-P.G., X.-D.W., Y.W. and L.-S.L. discussed the interpretation of results and wrote the manuscript. All authors discussed the results and commented on the manuscript.

## Competing interests
The authors declare no competing interests.
