## [Peer Review File · Nature Communications]

Visualizing the interfacial-layer-based epitaxial growth process toward organic core-shell architecturesREVIEWER COMMENTS

Reviewer #1 (Remarks to the Author):

This paper entitled “Visualizing the interfacial-layer-based epitaxial growth process toward organic core/shell architectures” is an exciting and significant work. In brief, in this work, the authors have firstly demonstrated a vivid visualization of the morphology evolution of organic heterostructures from barcoded to core/shell structures in the epitaxial growth process. The in-situ formation of an interfacial-layer in the hierarchical self-assembly route facilitates the lattice-matching epitaxial-growth process, which could suffer a low structural/chemical compatibility between the corresponding organic building blocks. Impressively, this epitaxial growth process could be generalized to other organic material systems, affording a universal avenue to the rational synthesis of desired organic heterostructures for advanced optoelectronics. Overall, the results are concrete and solid, and the manuscript is well-organized. Therefore, I'd like to recommend its publications in Nature Communications, provided the following comments are addressed.

1. There are limit data to confirm the charge-transfer (CT) character of these prepared organic CT cocrystals, such as the BGP-TFP (BTP) or BGP-TCNB (BTB) cocrystals.
2. There are some mistakes in the current manuscript. The Figure 1d is the SEM image of organic barcoded microwires, not the fluorescence microscope (FM). The similarly error was also found in Figure S13.
3. Why the PL isotropy of the red emission from the core/shell at the end part could confirm the that red emission of the core/shell part contributes to the doping of the red-emissive BTB?
4. What heterostructures will be found via a horizontal epitaxial growth after increasing the temperature of the stock solution more than 40°C?
5. There is not powerful data to verify the NIR emission of the prepared BCP-TCNQ cocrystals.
6. The authors illustrate that the interfacial-layer is favorable for the epitaxial growth in multistep seeded growth method, how about the universality in the sequential crystallisation process?
7. How about the stability of the organic barcoded heterostructured microwires at metastable state?
8. How about the influence of the concentration of the BTB and BTP stock solution in the hierarchical self-assembly of the organic barcoded microwires?
9. How about the universality of this mechanism in other organic cocrystals self-assembled based on halogen or hydrogen bonds?

Reviewer #2 (Remarks to the Author):

This manuscript demonstrated a comprehensive understanding of the hierarchical self-assembly of organic core/shell heterostructures with precise spatial configuration via a lattice-matching epitaxial-growth process based on the formation of an interfacial-layer. Impressively, the construction of organic core/shell heterostructures undergoing an epitaxial growth process of shell layers from tips to center surface of core part were powerfully affirmed by the formation of the metastable state of organic barcoded heterostructures with the periodic shells. Furthermore, the interfacial-layer-based epitaxial growth process with optimized lattice matching principle shows a charming universality for the precise synthesis of organic heterostructures, which supplies an avenue to adjust their physical/chemical properties for advanced applications. The novel view of the hierarchical self-assembly for organic heterostructures in this article is impressive, and it is expected to deliver a good impact to the community. We support the publication of this paper in Nature Communications after the correction of the paper shown below.

1. How long could these organic metastable heterostructure microwires retain the barcoded morphology?
2. This manuscript illustrated that the interfacial-layer is beneficial for lattice-matching epitaxial-growth process for forming the organic core/shell heterostructures based on the organic CT cocrystals. How about the universality in other kinds organic cocrystals?
3. Could be found the interfacial-layer in the segmented heterostructure?
4. How about the universality of the interfacial-layer-based epitaxial growth process in the physical vapor deposition (PVD) for forming organic heterostructures?
5. Compared with Figure 2e and 2f, why the length of the exposed core in the organic barcoded heterostructures prepared with epitaxial growth time of 15 min is shorter than that prepared with epitaxial growth time of 20 min.
6. Could the interfacial-layer-based epitaxial growth process be generalized to the organic systems with the building blocks of the single molecules for forming the organic heterostructures?

Reviewer #3 (Remarks to the Author):

M P Zhuo et al reported organic core-shell heterostructure with an interfacial layer. The author claims that the interfacial layer improves structural/chemical compatibility and reduces the barrier for the epitaxial growth of the shell layer. The organic barcoded heterostructure with periodic shell was established. The diameter, length, and number of periodic shells can be altered by changing the stoichiometric ratio (n) from 8:3 to 8:9, crystalline time and temperature. Further, they have designed some more examples of organic core-shell such as; BGP-OFN, BGP-TCIP, BGP-TCNB, BCP-TCIP, and BCP-TCNQ microwires.

However, the authors have published many papers on the same topic earlier and this presented work is incremental to their earlier works.

<https://doi.org/10.1002/anie.202117857>

<https://doi.org/10.1038/s41467-022-30870-y>

Science China Materials volume 66, pages733–739 (2023)

<https://doi.org/10.1021/acs.jpcllett.0c02293>

<https://doi.org/10.1038/s41467-019-11731-7>

And many more.

Although the work is prepared comprehensively, apart from the novelty part, it has several fundamental shortcomings that must be addressed carefully. Therefore, I cannot recommend the publication of this work in Nat Commun.

Comments

- It's better to include the chemical structure of used molecules in Figure 1.
- Page no. 5, line 145 change nano to micro in S9.
- Page no. 7, line 181 "Then, the BTP horizontally epitaxial-grows on the end surface" Rewrite the sentence.
- Page no. 7, line 201 "exposed core with a diameter of 2.0 μm " Why does the exposed core diameter first increase and then decrease?
- Page no. 8, line 235 The SEM images (Figure 3g) not g its h.
- Why epitaxial growth of the shell layer starts to grow from the tip of the crystal?
- Page no. 9, line 251 "Owing to the high defect density at the heterojunction, these prepared organic core/shell microwires show a strong red emission at the center part." This statement needs more explanation.
- Page no. 10, line 291 why barcoded number increase with the enhancement of temperature?
- Page no. 12, line 338/ Page no. 5, line 133/147 "Under the excitation of UV light," the author wrote "excitation of" multiple times which is incorrect, please correct the sentence.
- Fig 2 g, h, and i it is better to compare them with the same scale.

Reviewer #1 (Remarks to the Author):

This paper entitled “Visualizing the interfacial-layer-based epitaxial growth process toward organic core/shell architectures” is an exciting and significant work. In brief, in this work, the authors have firstly demonstrated a vivid visualization of the morphology evolution of organic heterostructures from barcoded to core/shell structures in the epitaxial growth process. The in-situ formation of an interfacial-layer in the hierarchical self-assembly route facilitates the lattice-matching epitaxial-growth process, which could suffer a low structural/chemical compatibility between the corresponding organic building blocks. Impressively, this epitaxial growth process could be generalized to other organic material systems, affording a universal avenue to the rational synthesis of desired organic heterostructures for advanced optoelectronics. Overall, the results are concrete and solid, and the manuscript is well-organized. Therefore, I'd like to recommend its publications in Nature Communications, provided the following comments are addressed.

1. There are limit data to confirm the charge-transfer (CT) character of these prepared organic CT cocrystals, such as the BGP-TFP (BTP) or BGP-TCNB (BTB) cocrystals.

Response: We greatly appreciate the reviewer's comments and feedback to help us further perfect our work. Furthermore, we also thank for reviewer's comments of our work that “Overall, the results are concrete and solid, and the manuscript is well-organized.” Furthermore, we have performance more experiments to confirm the charge-transfer (CT) character of these prepared organic BGP-TFP (BTP) or BGP-TCNB (BTB) cocrystals.

Solid-state ^{13}C NMR spectroscopy were reveals the chemical environment of the C atoms (*Angew. Chem. Int. Ed.* **2018**, *130*, 4027). According to solid-state ^{13}C NMR spectra (Figure R1a and R1b), after cocrystallization of BTB or BTP cocrystal, the chemical shift of BGP shows an obvious upfield shift from 125 to 127 or 128 ppm, indicating the increase of electron density (*Angew. Chem. Int. Ed.* **2020**, *60*, 6344). Meanwhile, the chemical shifts of TCNB at around 134 ppm and TFP at around 117 ppm exhibit downfield shifts. These observations demonstrate the π electron delocalization from TCNB or TFP to BGP molecules, implying the existence of intermolecular CT interactions (*Adv. Mater.* **2016**, *28*, 5954). The electron spin resonance (ESR) spectrum of BTP exhibits a strong resonance signal with g factor of 2.0037 (Figure R1c). This result suggests the existence of unpaired electrons in BTP, which arises from the CT process (*ACS Nano* **2022**, *16*, 15000). Furthermore, the stronger ESR signal of BTB compared to that of BTP suggests a stronger CT interaction between BGP and TCNB than that between BGP and TFP. As depicted in Figure R1d and R1e, the characteristic peak of $\text{C}\equiv\text{N}$ stretching vibrations at 2244 cm^{-1} in TCNB and 2248 cm^{-1} in TCNB is blue-shifted to 2242 cm^{-1} in BTB cocrystals and 2241 cm^{-1} in BTP cocrystals, respectively, suggesting a CT transition from BGP to TCNB or TFP (*Adv. Mater.* **2022**, *43*, 2107169). Compared to the constituent molecules, the BTP and BTB cocrystals exhibits a

broader red-shifted peak at 500 and 620 nm as verified in Figure R1f, which is corresponding to the CT transition from BGP to TFP or TCNB (*Angew. Chem. Int. Ed.* **2017**, *56*, 10352).

Figure R1. (a) and (b) Solid-state ^{13}C NMR spectra, (c) ESR spectra, (d and e) IR spectra, and (f) diffuse reflection absorption spectra of these of organic BTP and BTB cocrystals.

Revisions: The sentence of “These prepared BTP and BTB cocrystals presented the intermolecular CT transition from BGP to TCNB or TFP as verified by Fourier Transform Infrared (FT-IR) spectra, Solid-state ^{13}C NMR spectroscopy, electron spin resonance (ESR) spectra, and diffuse reflection absorption spectra in Figure S5.” was added on page 4, paragraph 2, lines 15-18 in the revised manuscript. The Figure R1 has been added in the supporting information as Figure S5.

2. There are some mistakes in the current manuscript. The Figure 1d is the SEM image of organic barcoded microwires, not the fluorescence microscope (FM). The similarly error was also found in Figure S13.

Response: Thanks for this ponderable suggestion. The result of that the obtained organic barcoded microwires show exposed core microwires with deep-red emission and a green emissive shell have been tautologically described at the impertinent Figure 1d. In fact, these organic barcoded microwires have a length of 55~70 μm with two thick shell layers at the tips and a thin core microwire at the center part, as verified by the scanning electron microscopy (SEM) image in Figure 1d. The similarly error in Figure S13 was also revised.

Revisions: The sentence of “As illustrated in Figure 1d, the obtained organic barcoded microwires show exposed core microwires with deep-red emission and a green emissive shell.” was revised in “As illustrated in Figure 1c, the obtained organic barcoded microwires show exposed core microwires with deep-red emission and an orange-red emissive shell excited by green light.” on page 5,

paragraph 2, lines 19-21 in the revised manuscript. The sentence of "... as evidenced by the FM and SEM images in Figure S11." was revised in "... as evidenced by the FM and SEM images in Figure S13." on page 7, paragraph 2, lines 9 in the revised manuscript.

3. Why the PL isotropy of the red emission from the core/shell at the end part could confirm the that red emission of the core/shell part contributes to the doping of the red-emissive BTB?

Response: Thanks for this instructive comment. Owing to anisotropic molecular packing and electronic coupling interactions, the organic crystals generally demonstrate the anisotropic optoelectronic properties (*Angew. Chem. Int. Ed.* **2018**, *130*, 11470; *Angew. Chem. Int. Ed.* **2019**, *132*, 4486). Typically, the PL intensity is at its maximum when the emission polarizer is parallel to the dipole moment of the emissive molecule (i.e., long molecular axis), and it is at its minimum when the emission polarizer is perpendicular to the dipole moment of the emissive molecule (*Adv. Funct. Mater.* **2005**, *15*, 17; *Adv. Mater.* **2010**, *22*, 3661). Therefore, the organic microcrystals generally display the PL anisotropic, which is attributed to the anisotropic molecular packing in the crystal structure (*J. Am. Chem. Soc.* **2019**, *141*, 6157; *Nat. Commun.* **2019**, *10*, 3839). In contrast, the random distribution of the doped molecules in the host crystal structure results in to an isotropy dipole moment of the emissive molecule, which could induce an isotropy emission property. Combining with the PL anisotropic of both red-emissive BTB and green-emissive BTP, the PL isotropy of the red emission from the core/shell at the end part could confirm the that red emission of the core/shell part contributes to the doping of the red-emissive BTB.

Revisions: The sentence of "As we well known that the organic microcrystals generally display the PL anisotropic, which is attributed to the anisotropic molecular packing in the crystal structure." was added on page 6, paragraph 1, lines 18-19 in the revised manuscript.

4. What heterostructures will be found via a horizontal epitaxial growth after increasing the temperature of the stock solution more than 40°C?

Response: Thanks for this ponderable suggestion. After increasing the temperature of the stock solution to 45 and 50°C, the organic barcoded microwires with more than four barcoded structures were unidentifiable as verified in Figure R2. Compared with the stock solution with the temperature of 40°C, the number of the barcoded block has an obvious enhancement. Furthermore, it is clearly found the additionally green-emissive BTP microcrystals, which is attributed to the high temperature for the supersaturation state, inducing an individual nucleation and growth for forming the BTP cocrystals. Owing to the rapid self-assemble process with the high temperature, the morphology of these prepared green-emissive BTP microcrystals are not regular. Due to the security consideration of the DCM solution in the high temperature more than 55°C, the experiment of the horizontal epitaxial growth with a temperature of the stock solution more than 55°C.

Figure R2. (a and b) FM images of the organic microcrystals obtained via a horizontal epitaxial growth after increasing the temperature of the stock solution more than 40°C. The scale bars are 20 μm. The corresponding excitation is UV light.

Revisions: The sentence of “After increasing the temperature of the stock solution to 45°C and 50°C, the unidentifiable obtained organic barcoded microwires with additionally green-emissive BTP microcrystals were obtained, which demonstrated an increase in the number of the barcoded blocks (Figure S23).” Was added on page 11, paragraph 1, lines 18-21 in the revised manuscript. The Figure R2 has been added in the supporting information as Figure S23.

5. There is not powerful data to verify the NIR emission of the prepared BCP-TCNQ cocrystals.

Response: Thanks for this ponderable suggestion. As shown in Figure R3a, the prepared BcP-TCNQ cocrystal shows broad absorption peaked at NIR region 760 nm and an obvious red-shift compared with that of the pure BcP or TCNQ crystals, which is corresponding to the CT transition from BcP to TCNQ (*Adv. Mater.* **2022**, *43*, 2107169). Owing to the broad and intense NIR absorption, the prepared BcP-TCNQ cocrystal emits the attractive NIR emission with a maximum PL peak at 776 nm, in contrast to the pure blue-emissive BcP with a maximum PL peak at 430 nm (Figure R3b).

Figure R3. (a) Absorption and (b) PL spectra of BcP-TCNQ cocrystal.

Revisions: The sentence of “Similarly, blue-emissive BCP-TCIP (BPP) microwires with a CIE of (0.20, 0.49) (Figure S22) and NIR-emissive BCP-TCNQ (BTQ) with a CIE of (0.73, 0.27) were successfully prepared via a solution self-assembly process.” was revised in “Similarly, blue-emissive BCP-TCIP (BPP) microwires with a CIE of (0.20, 0.49) (Figure S30) and NIR-emissive BCP-TCNQ (BTQ) with a CIE of (0.73, 0.27) (Figure S31) were successfully prepared via a

solution self-assembly process.” on page 15, paragraph 1, lines 2-4 in the revised manuscript. The Figure R3 has been added in the supporting information as Figure S31.

6. The authors illustrate that the interfacial-layer is favorable for the epitaxial growth in multistep seeded growth method, how about the universality in the sequential crystallisation process?

Response: Thanks for this ponderable suggestion. In order to investigate the universality in the sequential crystallization process, the organic triblock microwires based on the BTP and BTB blocks were controllably prepared in our previous work (*Nat. Commun.* **2021**, *12*, 2252). At the first state, the tips of the red-emissive BTB microwires demonstrate the yellow emission under the excitation of the UV band (Figure R4a). After changing the excitation from UV to green band, the no-emissive tips were clearly found as verified in Figure R4b. It is suggested that the yellow-emissive tips correspond to the integration green-emission of BTP tips and red-emission of BTB center part. Notably, there is a transient block between the red-emissive core and the yellow-emissive tips, which demonstrates the bright and weak red-emission under the excitation of UV and green band (Figure R4c and R4d), respectively. The high-efficiency energy transfers from BTP to BTB results in the bright red-emission at the UV band excitation. As well, the presence of the doped BTB in BTP induced a weak red emission of the limited amount BTB block, suggesting the formation of the interfacial-layer in the epitaxial growth in multistep seeded growth method. Furthermore, the yellow-emissive transient block between the red-emissive center part and the green-emissive tips (Figure R4e), which is the interfacial-layer. As a short conclusion, the interfacial-layer in the epitaxial growth has a universality in the sequential crystallisation process.

Figure R4. FM images of the organic triblock microwires prepared via a sequential crystallization process (a-d) at the first state and (e) final state. The excitations are (a, c and e) the UV band and (b and d) the green band (500-550 nm). The scale bars are (a, b, and e) 10 μm and (c and d) 2 μm , respectively.

Revisions: The sentence of “Furthermore, the formation of interfacial-layer in the epitaxial growth present an attractive universality in the sequential crystallization process (Figure S33), the physical vapor deposition (PVD) for forming organic heterostructures (Figure S34).” was added on page 15, paragraph 1, lines 27-29 and

16, paragraph 1, lines 1 in the revised manuscript. The Figure R4 has been added in the supporting information as Figure S33.

7. How about the stability of the organic barcoded heterostructure microwires at metastable state?

Response: Thanks for this ponderable suggestion. In order to prove the stability of organic barcoded heterostructure microwires at metastable state, the PL spectra, SEM and FM images of these organic barcoded heterostructures stored in air for 90 days are shown in Figure R4. After stored in air for 90 days, these as-prepared organic microwires still retain the barcoded morphology with an exposed deep-red-emissive core block and a shiny-red-emissive shell part under the excitation of UV light (Figure R4a). After changing the UV light to green light, these organic barcoded microwires displays an intense red emission as illustrated in Figure R4b. Furthermore, these organic barcoded microwires have a length of 70~80 μm with two thick shell layers at the tips and a thin core microwire with a diameter of $\sim 1\mu\text{m}$ at the center part (Figure R4c and R4d). As given in Figure R4e and R4f, the spatially resolved PL spectrum collected at location (A) at the exposed core part of the organic barcoded microwires by focusing a laser beam ($\lambda = 375\text{ nm}$) shows red emission with a PL peak at 600 nm, which is consistent with that of the BTB cocrystal. As well as, the additional green light with a PL peak at 515 nm emitted from the BTP shell emerges on the PL spectrum of location (B) at the core/shell part. Both the morphology and emission feature of the organic barcoded microwires after stored in air for 90 days are in consistent with those of the fresh sample as shown in Figure 2, which demonstrates their good stability.

Figure R5. (a and b) FM images of the typical organic barcoded heterostructures stored for 90 days with the excitation of (a) the UV band and (b) the green band (500–550 nm). The scale bars are 20 μm . (c and d) SEM images of the typical organic barcoded heterostructures stored for 90 days. The scale bars are 20 and 5 μm , respectively. (e) Bright-field and FM images of the individual organic barcoded heterostructure with a scale bar of 20 μm . (f) The Spatially resolved PL spectra collected from different locations marked in (e).

Revisions: The sentence of “Impressively, these prepared organic barcoded microwires still remain the barcoded structure and designated emission feature after stored in air for 90 days (Figure 12), which powerfully demonstrates their excellent stability.” was added on page 6, paragraph 1, lines 21-23 in the revised manuscript. The Figure R5 has been added in the supporting information as Figure S12.

8. How about the influence of the concentration of the BTB and BTP stock solution in the hierarchical self-assembly of the organic barcoded microwires?

Response: Thanks for this ponderable suggestion. The concentration of organic molecules is crucial influence for their controlled self-assemble process, which demonstrate an important influence for the molecular packing mode, morphology, and optoelectronic properties (*Angew. Chem. Int. Ed.* **2009**, *48*, 9121; *Angew. Chem. Int. Ed.* **2018**, *130*, 12653; *Matter* **2022**, *5*, 1520). After reducing the concentration of the BTB and BTP stock solution from 10 to 5 mmol/L, the defined organic barcoded microwires were successfully prepared, which have a distinct in the diameter of the exposed core wire as verified in Figure R6a and R6c. On contrast, the corresponding exposed core wire of the organic barcoded microwires prepared the concentration of the BTB and BTP stock solution of 15 mmol/L shows an obvious reducement in the diameter (Figure R6b and R6d). According to the abovementioned results, it is suggested that the concentration of the BTB and BTP stock solution in the hierarchical self-assembly has a necessary influence in the diameter of the exposed core wires in the organic barcoded heterostructures.

Figure R6. FM images of the organic barcoded heterostructures stored prepared with concentration of (a and c) 5 mmol/L and (b and d) 15 mmol/L for the BTB and BTP stock solution in the hierarchical self-assembly. The corresponding excitation are (a and b) the UV band and (c and d) the green band (500-550 nm), respectively. The scale bars are 10 μm .

Revisions: The sentence of “Furthermore, the diameter of the exposed wires in these prepared organic barcoded microwires could be controlled by finely adjusting the concentration of the BTB and BTP stock solution in the hierarchical self-assembly as elucidated in Figure S17.” was added on page 9, paragraph 1, lines 16-

19 in the revised manuscript. The Figure R6 has been added in the supporting information as Figure S17.

9. How about the universality of this mechanism in other organic cocrystals self-assembled based on halogen or hydrogen bonds?

Response: Thanks for this ponderable suggestion. The typical organic barcoded heterostructures were successfully prepared based on the organic halogen cocrystal of 4,4'-((1E,1'E)-(2,5-dimethoxy-1,4-phenylene)bis(ethene-2,1-diyl))dipyridine (DPEpe)- 1,4-diiidotetrafluorobenzene (F₄DIB) with strong green emission and the organic hydrogen cocrystal of DPEpe- 4-bromo-2,3,5,6-tetrafluorobenzoic acid (BrFTA) with bright red emission (*Nat. Commun.* **2019**, *10*, 3839). The red-emissive DPEpe-BrFTA microwires firstly self-assemble acting as the seeds. Then, the green-emissive DPEpe-F₄DIB cocrystals undergo a selective nucleation and the epitaxial growth on the surface of the seeded tips, resulting into the barcoded structures. It intensely confirms the desired the universality of this mechanism in other organic cocrystals self-assembled based on halogen or hydrogen bonds.

Revisions: The sentence of “Thus, the periodic shell layers epitaxially grown on the seeded surface based on an interfacial layer at the heterojunctions have been demonstrated to be a universal process to achieve barcoded and core/shell heterostructures.” was revised in “Thus, the periodic shell layers epitaxially grown on the seeded surface based on an interfacial layer at the heterojunctions have been demonstrated to be a universal process to achieve barcoded and core/shell heterostructures, as in the case of other kinds barcoded heterostructures based on organic halogen cocrystal of DPEpe-F₄DIB and organic hydrogen cocrystal of DPEpe-BrFTA.” on page 15, paragraph 1, lines 22-27 in the revised manuscript.

Reviewer #2 (Remarks to the Author):

This manuscript demonstrated a comprehensive understanding of the hierarchical self-assembly of organic core/shell heterostructures with precise spatial configuration via a lattice-matching epitaxial-growth process based on the formation of an interfacial-layer. Impressively, the construction of organic core/shell heterostructures undergoing an epitaxial growth process of shell layers from tips to center surface of core part were powerfully affirmed by the formation of the metastable state of organic barcoded heterostructures with the periodic shells. Furthermore, the interfacial-layer-based epitaxial growth process with optimized lattice matching principle shows a charming universality for the precise synthesis of organic heterostructures, which supplies an avenue to adjust their physical/chemical properties for advanced applications. The novel view of the hierarchical self-assembly for organic heterostructures in this article is impressive, and it is expected to deliver a good impact to the community. We support the publication of this paper in Nature Communications after the correction of the paper shown below.

1. How long could these organic metastable heterostructure microwires retain the barcoded morphology?

Response: We appreciate that the reviewer deems our results as “impressive” and our manuscript is expected to deliver a good impact to the community. In order to prove the stability of organic barcoded heterostructure microwires at metastable state, the PL spectra, SEM and FM images of these organic barcoded heterostructures stored in air for 90 days are shown in Figure R7. After stored in air for 90 days, these as-prepared organic microwires still retain the barcoded morphology with an exposed deep-red-emissive core block and a shiny-red-emissive shell part under the excitation of UV light (Figure R7a). After changing the UV light to green light, these organic barcoded microwires displays an intense red emission as illustrated in Figure R7b. Furthermore, these organic barcoded microwires have a length of 70~80 μm with two thick shell layers at the tips and a thin core microwire with a diameter of $\sim 1\mu\text{m}$ at the center part (Figure R7c and R7d). As given in Figure R6e and R6f, the spatially resolved PL spectrum collected at location (A) at the exposed core part of the organic barcoded microwires by focusing a laser beam ($\lambda = 375\text{ nm}$) shows red emission with a PL peak at 600 nm, which is consistent with that of the BTB cocrystal. As well, the additional green light with a PL peak at 515 nm emitted from the BTP shell emerges on the PL spectrum of location (B) at the core/shell part. Both the morphology and emission feature of the organic barcoded microwires after stored in air for 90 days are in consistent with those of the fresh sample as shown in Figure 2, which demonstrates their good stability. It is also indicated that these organic metastable heterostructure microwires could retain the barcoded morphology less than 90 days.

Figure R7. (a and b) FM images of the typical organic barcoded heterostructures stored for 90 days with the excitation of (a) the UV band and (b) the green band (500–550 nm). The scale bars are 20 μm . (c and d) SEM images of the typical organic barcoded heterostructures stored for 90 days. The scale bars are 20 and 5 μm , respectively. (e) Bright-field and FM images of the individual organic barcoded heterostructure with a scale bar of 20 μm . (f) The Spatially resolved PL spectra collected from different locations marked in (e).

Revisions: The sentence of “Impressively, these prepared organic barcoded microwires still remain the barcoded structure and designated emission feature after stored in air for 90 days (Figure 12), which powerfully demonstrates their excellent

stability.” was added on page 6, paragraph 1, lines 21-23 in the revised manuscript. The Figure R7 has been added in the supporting information as Figure S12.

2. This manuscript illustrated that the interfacial-layer is beneficial for lattice-matching epitaxial-growth process for forming the organic core/shell heterostructures based on the organic CT cocrystals. How about the universality in other kinds organic cocrystals?

Response: Thanks for this ponderable suggestion. The typical organic barcoded heterostructures were successfully prepared based on the organic halogen cocrystal of 4,4'-((1*E*,1'*E*)-(2,5-dimethoxy-1,4-phenylene)bis(ethene-2,1-diyl))dipyridine (DPEpe)- 1,4-diodotetrafluorobenzene (F₄DIB) with strong green emission and the organic hydrogen cocrystal of DPEpe- 4-bromo-2,3,5,6-tetrafluorobenzoic acid (BrFTA) with bright red emission (*Nat. Commun.* **2019**, *10*, 3839). The red-emissive DPEpe-BrFTA microwires firstly self-assemble acting as the seeds. Then, the green-emissive DPEpe-F₄DIB cocrystals undergo a selective nucleation and the epitaxial growth on the surface of the seeded tips, resulting into the barcoded structures. It intensely confirms the desired the universality of this mechanism in other organic cocrystals self-assembled based on halogen or hydrogen bonds.

Revisions: Revisions: The sentence of “Thus, the periodic shell layers epitaxially grown on the seeded surface based on an interfacial layer at the heterojunctions have been demonstrated to be a universal process to achieve barcoded and core/shell heterostructures.” was revised in “Thus, the periodic shell layers epitaxially grown on the seeded surface based on an interfacial layer at the heterojunctions have been demonstrated to be a universal process to achieve barcoded and core/shell heterostructures, as in the case of other kinds barcoded heterostructures based on organic halogen cocrystal of DPEpe-F₄DIB and organic hydrogen cocrystal of DPEpe-BrFTA.” on page 15, paragraph 1, lines 22-27 in the revised manuscript.

3. Could be found the interfacial-layer in the segmented heterostructure?

Response: Thanks for this ponderable suggestion. In order to investigate the universality in the sequential crystallization process, the organic triblock microwires based on the BTP and BTB blocks were controllably prepared in our previous work (*Nat. Commun.* **2021**, *12*, 2252). At the first state, the tips of the red-emissive BTB microwires demonstrate the yellow emission under the excitation of the UV band (Figure R4a). After changing the excitation from UV to green band, the no-emissive tips were clearly found as verified in Figure R4b. It is suggested that the yellow-emissive tips correspond to the integration green-emission of BTP tips and red-emission of BTB center part. Notably, there is a transient block between the red-emissive core and the yellow-emissive tips, which demonstrates the bright and weak red-emission under the excitation of UV and green band (Figure R4c and R4d), respectively. The high-efficiency energy transfers from BTP to BTB results in the bright red-emission at the UV band excitation. As well, the presence of the doped BTB in BTP induced a weak red emission of the limited

amount BTB block, suggesting the formation of the interfacial-layer in the epitaxial growth in multistep seeded growth method. Furthermore, the yellow-emissive transient block between the red-emissive center part and the green-emissive tips (Figure R4e), which is the interfacial-layer. As a short conclusion, the interfacial-layer in the epitaxial growth has a universality in the sequential crystallisation process.

Figure R8. FM images of the organic triblock microwires prepared via a sequential crystallization process (a-d) at the first state and (e) final state. The excitation are (a, c and e) the UV band and (b and d) the green band (500-550 nm). The scale bars are (a, b, and e) 10 μm and (c and d) 2 μm , respectively.

Revisions: The sentence of “Furthermore, the formation of interfacial-layer in the epitaxial growth present an attractive universality in the sequential crystallization process (Figure S33), the physical vapor deposition (PVD) for forming organic heterostructures (Figure S34).” was added on page 15, paragraph 1, lines 27-29 and 16, paragraph 1, lines 1 in the revised manuscript. The Figure R8 has been added in the supporting information as Figure S33.

4. How about the universality of the interfacial-layer-based epitaxial growth process in the physical vapor deposition (PVD) for forming organic heterostructures?

Response: Thanks for your valuable suggestion. In order to investigate the universality of the interfacial-layer-based epitaxial growth process in the physical vapor deposition (PVD) for forming organic heterostructures, the previous work (*ACS Nano* **2022**, *16*, 3290) was repeated. As shown in Figure R8a, the organic triblock microwires with the typical red-green-red emission were successfully prepared via the same condition. When excited by blue and green light, the red-emission of the tips changes into the no-emission and red emission, as well, the green emission of the end parts changes into green emission and no-emission as verified by Figure R8b and R8c. Notably, there is an obvious distance between the red and green emission as given in Figure R8d-R8f, which is corresponding to the interfacial-layer. Therefore, the interfacial-layer-based epitaxial growth process demonstrated the universality in the physical vapor deposition (PVD) for forming organic heterostructures.

Figure R8. FM images of organic barcoded microwires prepared by physical vapor deposition (PVD) method. Excitation are (a, and d) the UV band, (b, and e) blue light, and (c and f) green light, respectively. The scale bars are (a-c) 20 μm and (d-f) 5 μm .

Revisions: The sentence of “Furthermore, the formation of interfacial-layer in the epitaxial growth present an attractive universality in the sequential crystallization process (Figure S33), the physical vapor deposition (PVD) for forming organic heterostructures (Figure S34).” was added on page 15, paragraph 1, lines 27-29 and 16, paragraph 1, lines 1 in the revised manuscript. The Figure R8 has been added in the supporting information as Figure S34.

5. Compared with Figure 2e and 2f, why the length of the exposed core in the organic barcoded heterostructures prepared with epitaxial growth time of 15 min is shorter than that prepared with epitaxial growth time of 20 min.

Response: Thanks for your valuable suggestion. It is our mistake that the selected organic barcoded microwires with epitaxial growth time of 15 min was not appropriate. As shown in Figure R9a, the length of the exposed core in the organic barcoded heterostructures prepared with epitaxial growth time of 15 min is not shorter than that prepared with epitaxial growth time of 20 min. The diameter of the exposed core and complete core/shell parts are 2.1 and 4.2 μm , respectively (Figure R9b). The ratio between the core/shell block and the exposed core block is 0.57, which is consistent with that prepared with epitaxial growth time of 20 min.

Figure R9. (a and b) SEM images of the organic barcoded heterostructures prepared with epitaxial growth time of 15 min.

Revisions: The Figure 2e were replaced by the Figure R9b in the revised manuscript.

6. Could the interfacial-layer-based epitaxial growth process be generalized to the organic systems with the building blocks of the single molecules for forming the organic heterostructures?

Response: Thanks for your valuable suggestion. The interfacial-layer-based epitaxial growth process via a doping process could be generalized to the organic systems with the building blocks of the single molecules for forming the organic heterostructures, which was developed by our group and coworkers (*Nano Lett.* **2017**, *17*, 695). Typically, an efficient doping step for coronene and perylene domains enables their perfect lattice matching, which facilitates facet-selective epitaxial growth of perylene domains on both the tips and the side surfaces of the preformed coronene seeded microwires by manipulating the growth pathways of the two pairs of materials under a sequential crystallization process. In that work, we just inferred that an efficient doping process at the heterojunction for coronene and perylene domains may be involved in the present coassembly system. At that time, we could not realize the importance of the interfacial-layer forming in the epitaxial growth via an efficient doping for facially and controllably preparing the organic heterostructures. This novel doping process in the formation of organic heterostructures between the different building blocks of the single molecule were also found, such the organic branched heterostructures containing 2,5,8,11-tetra-tert-butylperylene and 1,4-dimethoxy-2,5-di(E)-bis(2-methylstyryl) benzene or 2,2'-((1E,1'E)-1,4-phenylenebis(ethene-2,1diyl))-dibenzonitrile (*ACS Mater. Lett.* **2020**, *2*, 658). Furthermore, this doping process for the formation of organic heterostructures was also found discovered by other groups (*Mater. Horiz.* **2019**, *6*, 984; *Angew. Chem. Int. Ed.* **2021**, *133*, 27252; *J. Mater. Chem. C* **2021**, *9*, 489; *Adv. Optical Mater.* **2022**, *11*, 2200776), suggesting a high universality. Depended on our deeply study on the rational design and fine synthesis of the organic heterostructures, we firstly have a comprehensive understanding that the interfacial-layer-based epitaxial growth process is beneficial for high chemical/structural compatibility, which is conducive to the formation of the organic heterostructures.

Revisions: The sentence of “Therefore, the controlled supersaturation (*S*) of BTP and BTB was achieved via finely tuning the volume ratio between methanol and hexamethylene” is revised in “ Therefore, the controlled supersaturation (*S*) of BTP and BTB was achieved via finely tuning the volume ratio between methanol and hexamethylene, determining the absence of the BTB doping process in the heterogeneous-nucleation epitaxial- growth of the BTP shell, which is consistent with the previous works based on the building blocks of the single molecules.³¹⁻³³” on page 5, paragraph 2, lines 7-11 in the revised manuscript. The corresponding Refers 31-33 is added in the revised manuscript.

Reviewer #3 (Remarks to the Author):

M. P. Zhuo et al reported organic core-shell heterostructure with an interfacial layer. The author claims that the interfacial layer improves structural/chemical compatibility and reduces the barrier for the epitaxial growth of the shell layer. The organic barcoded heterostructure with periodic shell was established. The diameter, length, and number of periodic shells can be altered by changing the stoichiometric ratio (n) from 8:3 to 8:9, crystalline time and temperature. Further, they have designed some more examples of organic core-shell such as; BGP-OFN, BGP-TCIP, BGP-TCNB, BCP-TCIP, and BCP-TCNQ microwires.

However, the authors have published many papers on the same topic earlier and this presented work is incremental to their earlier works.

<https://doi.org/10.1002/anie.202117857>

<https://doi.org/10.1038/s41467-022-30870-y>

Science China Materials volume 66, pages733–739 (2023)

<https://doi.org/10.1021/acs.jpcelett.0c02293>

<https://doi.org/10.1038/s41467-019-11731-7>

And many more.

Although the work is prepared comprehensively, apart from the novelty part, it has several fundamental shortcomings that must be addressed carefully. Therefore, I cannot recommend the publication of this work in *Nat. Commun.*

Response: We greatly appreciate the reviewer's comments and feedback to help us further perfect our work. Furthermore, we also thank for reviewer's comments of our current work that "the work is prepared comprehensively", which is like with the valuable comments from other two reviewers' comments. We have carefully answered reviewer's comments and provided corresponding explanations in the revised manuscript. The detailed point-by-point responses of the reviewer's comments are as follows. We believe that the revised version is satisfactory for the publication in *Nat. Commun.*, and sincerely hope the reviewer reconsider our manuscript.

Significantly, we want to take this chance to clarify the novelty and the significance of our current work as follows. As well known, the rational construction of the organic low-dimensional heterostructures (OLDHs) has gained impressive attention in material chemistry as promising candidates to meet the practical nanotechnological requirements for the advanced optoelectronics devices with high-performance (*Science* **2004**, *306*, 1009; *Chem. Rev.* **2016**, *116*, 2775). Up to date, we have really published a series of works on the rational design/synthesis of various types of traditional OLDHs with accurate spatial organization, including the core/shell, branched, and segmented heterostructures via finely regulating the noncovalent interactions or multicomponent supersaturation for the tunable and orderly epitaxial-growth process. The representative cases include tuning hydrogen and halogen bonds for triblock heterostructure (*Nat. Commun.* **2019**, *10*, 3839), tuning charge-transfer (CT) and π - π interactions for core/shell heterostructure (*Adv. Mater.* **2021**, *33*, 2102719) and branched heterostructure (*J. Phys. Chem. Lett.* **2020**, *11*, 7517). **In this work, we**

firstly and successfully found the intermediate growth process of interfacial-layer between the seeded microwire and the shell layer in the hierarchical self-assembly process of organic core/shell microwires. Significantly, the corresponding growth mechanism was studied systematically via the visualization of morphology evolution for the final formation of organic core/shell heterostructures, which demonstrates the comprehensively understanding the accurate growth processes of organic core/shell heterostructures. Impressively, our discovered interfacial-layer-based epitaxial growth process was employed to finely the organic barcoded microwire arrays, which could further clarify the novelty of this current work beyond to our previous works. The detail research result was as follows:

The organic barcoded microwire arrays were successfully and firstly prepared via a capillary-bridge confined assembly method based on interfacial-layer-based epitaxial growth process as illustrate in Figure R10a (*Adv. Mater.* **2019**, *31*, 1807880; *Matter*, **2020**, *2*, 1233). Following the BTP crystal heteronucleation and epitaxial growth with the BTB doped process on the surface of the BTB microwire arrays, the organic barcoded microwire arrays with the red-emissive shell layer on the surface of a slender core microwire were successfully prepared, which present the typical barcoded heterostructures similar to that obtained by the solution self-assemble method. Due to being at (III) state in the kinetics of solute supersaturation (Figure R10g), the additional BTP stock solution could not affect the morphology of the BTB microwire arrays. Furthermore, the residuary BTB component firstly will dope in the BTP to form the interfacial layer. Notably, the contour map of the barcoded distance indicates the synergistic effect of temperature and concentration in second self-assembly process on the regulation of density and location of the periodic shell layer (Figure R10f). The distance between the adjacent shell layer from 3 to 50 μm and the density of the periodic shell layer could be precisely controlled by tuning the system temperature and the concentration of BTP stock solution (Figure R10b-R10e). The enhanced temperature and increased concentration led to more heteronucleation of the BTP on the center surface of the BTB core microwires due to the reduced energy barrier, forming organic barcoded microwires with dense periodic shells. In conclusion, multiple levels of confined assembly method could fabricate highly ordered organic barcoded microwires, which is still an undeveloped domain.

The highly ordered organic micro/nano-crystal arrays demonstrate the fascinating on-demand photons/electrons transport features, which present an enticing prospect for the large-scale and integrated optoelectronic application (*J. Am. Chem. Soc.* **2003**, *125*, 4728; *Adv. Mater.* **2015**, *27*, 7305). Combing with the unique exciton conversion and photon propagation at the heterojunction (*Angew. Chem. Int. Ed.* **2022**, *134*, 202117857; *Adv. Mater.* **2022**, *35*, 2206272), these prepared organic barcoded microwire arrays provide great advantages in the field of precisely regulating photon transmission, which hold great promise toward both fundamental types of research and applications of organic photonic circuits.

Figure R10. (a) Schematic representation (top) and fluorescence observation of the dewetting process of capillary-bridge assembly system. (b-c) Morphology images of organic barcoded microwires arrays based on the temperature of a capillary-bridge confined assembly method: (b) 30, (c) 35, and (d) 40 °C. The scale bars are 20 μm. (e) The distribution of the core microwire distance between the two shell layer. (f) Contour plots of the core microwire distance between the shell layer assembled by different concentration of BTP stock solution at different temperatures. (g) Kinetics of solute supersaturation including (I) supersaturation, (II) nucleation, and (III) growth. Precursor conversion reactions that limit the crystallization determine the temporal evolution of monomer concentration as well as the steady state supersaturation during the growth phase.

In summary, this work gives the first discovery on the interfacial-layer-based epitaxial growth process with optimized lattice matching principle for the precise synthesis and patterning of OLDH arrays. We hope the revised version can address the reviewer’s comments, and we believe this work will deliver a good impact to the community.

Revisions: The Figure R10 were added as Figure 5 in the revised manuscript. And the corresponding description paragraph also added on page 12 and 14 revised manuscript.

Detailed point-by-point responses:

1. It’s better to include the chemical structure of used molecules in Figure 1.

Response: Thanks for your ponderable suggestion. Driving by the charge-transfer (CT) interaction, benzo[ghi]perylene (BGP) and 1,2,4,5-Benzenetetrinitrile

(TCNB) self-assemble to BTB cocrystal, as well as BGP and tetrafluoroterephthalonitrile (TFP) self-assemble to BTP cocrystal. The molecular structures of BGP, TCNB, and TFP were shown in Figure R10.

Figure R11. The molecular structures of (a) benzo[ghi]perylene, (b) 1,2,4,5-Benzenetetraniitrile, and (c) tetrafluoroterephthalonitrile.

Revisions: The sentence of “The polycyclic aromatic hydrocarbon (PAH) of benzo[ghi]perylene (BGP) demonstrates a planar aromatic π -conjugated structure and a strong electron-donating ability...” is revised to “The polycyclic aromatic hydrocarbon (PAH) of benzo[ghi]perylene (BGP) demonstrates a planar aromatic π -conjugated structure and a strong electron-donating ability (Figure 1a)...” on page 4, paragraph 2, lines 1-5 in the revised manuscript. The chemical structure of used molecules in Figure R11 is added in Figure 1a in revised manuscript.

2. Page no. 5, line 145 change nano to micro in S9.

Response: Thanks for your valuable suggestion. The word of “nanowires” is revised into “microwires” on no. 5, line 145 in S9. Furthermore, the other “nanowires” is also revised into “microwires”.

Revisions: The word of “nanowires” is revised into “microwires” in Figure S9. Furthermore, the other “nanowires” is also revised into “microwires” in the revised manuscript and supporting information.

3. Page no. 7, line 181 “Then, the BTP horizontally epitaxial-grows on the end surface” Rewrite the sentence.

Response: We are very sorry for mistake. The rewrite the sentence of “Then, the BTP horizontally epitaxial-grows on the end surface” is deleted on Page 7, line 181 in the revised manuscript

Revisions: The rewrite the sentence of “Then, the BTP horizontally epitaxial-grows on the end surface” is deleted on Page 7, line 181 in the revised manuscript.

4. Page no. 7, line 201 “exposed core with a diameter of 2.0 μm ” Why does the exposed core diameter first increase and then decrease?

Response: Thanks for your valuable suggestion. The diameter of the exposed core corresponding to the organic barcoded microwires prepared with epitaxial growth times of 10, 15, and 20 min are 2.1, 2.2, and 2.0 μm , respectively, presenting an average diameter of 2.0 μm . It suggests the exposed core diameter of the organic barcoded microwires prepared with epitaxial growth times first increase from 2.1 to 2.2 μm , and then decrease to 2.0 μm is normal. Theoretically, the exposed core of these organic barcoded microwires prepared with different epitaxial growth

times should remain the same diameter. In fact, it is impossibility to achieve the duplicate same experiment condition expect the epitaxial growth time for accurately prepare the completely same sample, which could induce some tiny difference in the micro/nanostructures, such as the tiny difference in the diameter of the exposed core.

Revisions: The sentence of "...exposed core microwires display an unaltered diameter of $\sim 2.0 \mu\text{m}$ during the whole epitaxial growth process (Figure 2j)." is revised to "...exposed core microwires display an unaltered diameter of $\sim 2.0 \mu\text{m}$ during the whole epitaxial growth process (Figure 2j), which is agreed with the representative units as shown in Figure 2g-2i." on page 8, paragraph 1, lines 5-7 in the revised manuscript.

5. Page no. 8, The SEM images (Figure 3g) not g its h.

Response: Thanks for your ponderable suggestion. The SEM images as described on page no. 8 line 235 is corresponding to Figure 3h, not Figure 3g. We have revised this mistake.

Revisions: The sentence of "The SEM images (Figure 3g) clearly illustrate that these prepared organic heterostructure microwires exhibit a smooth surface and a clear heterojunction." is revised to "The SEM images (Figure 3h) clearly illustrate that these prepared organic heterostructure microwires exhibit a smooth surface and a clear heterojunction." on page 9, paragraph 1, lines 10-12 in the revised manuscript.

6. Why epitaxial growth of the shell layer starts to grow from the tip of the crystal?

Response: Thanks for your valuable suggestion. The rapid self-assembly of BTB seeded along the [001] direction for forming the BTB microwires induces the abundant defect with a high surface energy at their unstable tips (*Science* **2004**, 304, 1787), which is favorable for the heteronucleation and epitaxial growth of organic heterostructures (*Adv. Mater.* **2012**, 24, 2332; *J. Am. Chem. Soc.* **2013**, 135, 3744). Furthermore, the surface free energy $E_{(\text{hkl})\text{s}}^{\text{surf}}$ of the exposed crystal planes in the growth morphology of BTB microwires follow the order: $|E_{(011)\text{s}}^{\text{surf}}| = |E_{(0-11)\text{s}}^{\text{surf}}|$ (-111.71 kcal/mol) $> |E_{(110)\text{s}}^{\text{surf}}| = |E_{(1-10)\text{s}}^{\text{surf}}|$ (-79.82 kcal/mol) $> |E_{(020)\text{s}}^{\text{surf}}|$ (-34.29 kcal/mol). The crystal planes of (011) and (0-11) at the tips of the BTB microwires demonstrated the high surface free energy compared with other crystal planes. The surface-interface energy balance considerations would also prefer to the selectively heteronucleation of the BTP on the tips (*J. Am. Chem. Soc.* **2010**, 132, 2437). Combing with high surface free energy and defect density at the unstable tips of BTB seeded microwires, the epitaxial growth of the shell layer preferentially starts to grow from the tip of the crystal for forming the organic heterostructures, as in the case of organic dumbbell-like heterostructures comprising coronene and perylene (*Nano Lett.* **2017**, 17, 695).

Revisions: The sentence of "After nucleation and growth at the early stage, BTB self-assembled into seeded microwires." is revised to "After nucleation and growth

at the early stage, BTB self-assembled into seeded microwires with high defect density at tips owing to the rapid self-assembly.” on page 11, paragraph 2, lines 3-5 in the revised manuscript. The sentence of “Because of the highest surface free energy, BTB is favorable to selectively heteronucleate on the crystal planes of (011)s and (0-11)s at the tips (Figure S20a and S20b).” is revised to “In order to reduce the highest surface free energy for surface-interface energy balance, BTB is favorable to selectively heteronucleate on the crystal planes of (011)s and (0-11)s at the tips (Figure S24a and S24b).” on page 12, paragraph 1, lines 5-7 in the revised manuscript.

7. Page no. 9, line 251 “Owing to the high defect density at the heterojunction, these prepared organic core/shell microwires show a strong red emission at the center part.” This statement needs more explanation.

Response: Thanks for this instructive comment. The smooth surface and high crystallinity minimized the optical loss caused by scattering, showing the bright emission spots at both tips and the weaker emission from bodies in the 1D microstructures (*Adv. Mater.* **2011**, *23*, 1380; *Angew. Chem. Int. Ed.* **2013**, *52*, 8713). The high defect density at the heterojunction of the prepared organic core/shell microwires at the center part is favorable for the increased scattering process, which induces an enhanced red emission in contrast with the weak emission at body, as in the case of the intense emissive heterojunction between the branched and track parts (*J. Am. Chem. Soc.* **2014**, *136*, 2382). In fact, the strong red emission at the heterojunction is attributed to the optical loss caused by the defect.

Revisions: The sentence of “Owing to the high defect density at the heterojunction, these prepared organic core/shell microwires show a strong red emission at the center part.” is revised to “Owing to the high defect density at the heterojunction inducing the increased optical scattering, these prepared organic core/shell microwires show a strong red emission at the center part.” on page 9, paragraph 2, lines 9 and page 10, paragraph 1, lines 1-2 in the revised manuscript.

8. Page no. 10, line 291 why barcoded number increase with the enhancement of temperature?

Response: Thanks for this instructive comment. It is well known that the high temperature could increase the surface energy of the nanomaterials. The periodically spaced particles benefit to reducing the total surface energy compared to the original one-dimensional material (*Nat. Nanotechnol.* **2015**, *10*, 345). Therefore, the periodically spaced shells on a one-dimensional BTB substrate reduce the total surface energy compared to a uniform-diameter nanowire of equivalent volume, thermodynamically driving growth of periodic shells over uniform-diameter shells, demonstrating the increased barcoded number. Furthermore, the increased temperature could induce a high evaporation rate of the organic solvent, resulting into the high supersaturation concentration of BTP cocrystals for the rapid self-assemble process. It is suggested that BTP could reach

supersaturation instantly during the crystallization process under the enhanced temperature. Combined with the low lattice mismatch ratio (f) of 0.8% ($d_{\text{BTP}}^{(100)} = 9.15 \text{ \AA} \approx d_{\text{BTP}}^{(110)} = 9.22 \text{ \AA}$), their high chemical and structural compatibility facilitates facet-selective heteronucleation and horizontally epitaxial growth of the BTP shell layer on the surface of BTB seeds for the construction of organic heterostructures. During this process, BTP would not have enough time to seek a topological configuration with minimized overall surface and interfacial energy, thereby driving the epitaxial growth of a large amount of BTB cocrystal nucleus containing BTB on side surface of BTB seed wires with larger surface to volume ratio, which is agreed with the previous works (*Angew. Chem. Int. Ed.* **2010**, 49, 4878; *Nano Today* **2010**, 5, 449; *Nano Lett.* **2017**, 17, 695).

Revisions: The sentence of “Furthermore, the increased temperature could induce a high evaporation rate of the organic solvent, resulting into the high supersaturation concentration of BTP cocrystals for the rapid self-assemble process. During this process, there are not enough time the subsequently formed BTP nucleus to choose a topological configuration with minimized overall surface and interfacial energy, inducing the epitaxial growth of abundant BTP cocrystal containing BTB on side surface of BTB seeded wires with larger surface to volume ratio, showing an increased shell number.” were added on page 12, paragraph 1, lines 16-23 in the revised manuscript.

9. Page no. 12, line 338/ Page no. 5, line 133/147 “Under the excitation of UV light,” the author wrote “excitation of” multiple times which is incorrect, please correct the sentence.

Response: Thanks for your valuable suggestion. The “excitation of” is incorrect. The same mistakes were revised in the revised manuscript.

Revisions: The phrase of “Under the excitation of UV light” is revised into “Upon excitation with UV light” on page 5, paragraph 2, lines 14; page 9, paragraph 1, lines 7; and page 15, paragraph 1, lines 8 in the revised manuscript.

10. Fig 2 g, h, and i it is better to compare them with the same scale.

Response: Thanks for your ponderable suggestion. The scales of Fig 2 g, h, and i are revised into the same as verified in Figure R12.

Figure R12. SEM images of organic barcoded microwires prepared with epitaxial growth times of (a) 10 min, (b) 15 min, and (c) 20 min. The scale bars are 5 μm .

Revisions: The Figure 2g, 2h, and 2i are replaced by Figure R12.

REVIEWER COMMENTS

Reviewer #1 (Remarks to the Author):

All my questions have been addressed in the revision, which could be acceptable.

Reviewer #2 (Remarks to the Author):

The authors have addressed all my concerns and it can be accepted for publication at the present form.

Reviewer #3 (Remarks to the Author):

The scales in Fig 2 g, h, and i are not correct, visibly. A quick check by using the scale bar in each figure and the measurement mentioned in the figures as labels reveals the mismatch. Pls, check.

Further, in Fig 2 d,e,f the scale bars are missing.

Additionally, I do not understand the logic of comparing epitaxial growth w.r.t time in different areas of the sample. Obviously, the rate of growth of crystals will not be uniform on the substrate, if this is not the case, once should expect same sized crystal after the self-assembly.

Page 4, line 113: "These prepared BTP and BTBa in Figure S5." This sentence is not complete. Further, use the word spectroscopy or spectra - do not mix both.

Fig 3. g,h,l,n,- The ratios of BTB and BTP are not mentioned.

Reviewer #1 (Remarks to the Author):

All my questions have been addressed in the revision, which could be acceptable.

Response: Thank you very much for recommending our work for publication and thank you again for your efforts to improve this work.

Reviewer #2 (Remarks to the Author):

The authors have addressed all my concerns and it can be accepted for publication at the present form.

Response: We appreciate your efforts to improve our work and recommend our work for publication.

Reviewer #3 (Remarks to the Author):

The scales in Fig 2 g, h, and i are not correct, visibly. A quick check by using the scale bar in each figure and the measurement mentioned in the figures as labels reveals the mismatch. Pls, check.

Further, in Fig 2 d,e,f the scale bars are missing.

Response: We greatly appreciate the reviewer's comments and feedback to help us further perfect our work. We are very sorry for our careless about the mismatch in Fig 2g-2i, as well as the missing scale bars in Figure 2d-2f. Firstly, the scale bars (missing in Figure 2d-2f) of 20 μm were added as shown in Figure R1a-R1c.

Figure R1. SEM images of organic barcoded microwires prepared with epitaxial growth times of (a) 10 min, (b) 15 min, and (c) 20 min. The scale bars are 20.

The SEM images in the previous manuscript is accurate as shown in Figure 2g-2i. The SEM image of Figure 2g and 2h was not zoom in the same proportion with the scale bar from 2 to 5 μm, leading to a mismatch in the revision manuscript. This mistake was perfected as shown in Figure R2d-R2f. Furthermore, the corresponding diameter of the shell layer in Figure R2e is 4.8 μm. The SEM image for the Figure 2i was simply changed without the consideration on the difference in the diameters of the core and shell parts as present in the previously revised manuscript, inducing a terrible mismatch. The accurate high-resolution SEM of organic barcoded microwires with an epitaxial growth times of 20 min was applied to replace the inapposite SME image in Figure 2i, which is presented in Figure R2f.

Figure R2. SEM images of organic barcoded microwires prepared with the epitaxial-growth time of (a) 10 min, (b) 15 min, and (c) 20 min, respectively. The scale bars are 5 μm .

Revisions: The sentence ‘..... while the diameter of the complete core/shell parts increases to 4.3 μm (Figure 2h).’ is revised to be ‘..... while the diameter of the complete core/shell parts increases to 4.8 μm (Figure 2h).’ on page 8, paragraph 1, lines 4-5 in the revised manuscript. Figure 2d-2i was replaced by Figure R1 in revision manuscript.

Additionally, I do not understand the logic of comparing epitaxial growth w.r.t time in different areas of the sample. Obviously, the rate of growth of crystals will not be uniform on the substrate, if this is not the case, once should expect same sized crystal after the self-assembly.

Response: Thanks for your ponderable suggestion. In situ visualization is widely applied to have a more direct and specific inspection of the self-assembly mechanism in sequential self-assembled process of the organic heterostructure architectures (*J. Am. Chem. Soc.* **2018**, *140*, 4269). It is well known that the organic molecular self-assembly is a dynamic process, difficult manipulation and the impressionable by the temperature, air pressure, shake, and other environmental conditions by the temperature, air pressure, shake, and other environmental conditions. Therefore, it is still a challenge to achieve the in-situ visualization of hierarchical self-assembly of the organic heterostructures. During the fluorescence microscopic imaging process, the light source would instantly increase the solution temperature and the solvent volatilization, resulting in the organic microstructures without desired spatial organization of barcoded heterostructure. Impressively, the corresponding growth rate of crystals will not be uniform on the substrate. It is suggested that the in-situ visualization of epitaxial growth process of the same organic barcoded microwires could be unsuccessfully achieved in our current work. As shown in core and shell diameter distributions of the organic barcoded microwires (Figure R3), the epitaxial-growth time plays a crucial role for the morphology of these prepared organic barcoded microwires. The core and shell diameter of the selective organic barcoded microwires as shown in Figure R2d-R2f (Figure 2g-2i) are consist with these time-based statistical distributions. Therefore, the time-based statistical fluorescence microscopy (FM) and SEM images were rationally performed and applied to investigate the hierarchical self-assembly mechanism of the organic barcoded microwires in the current work.

Figure R3. The distribution of the core and shell dimeter corresponding to the organic barcoded microwires prepared with the epitaxial-growth time of (a) 10 min, (b) 15 min, and (c) 20 min, respectively.

Revisions: The sentence ‘Due to the rapidly increased solution temperature and solvent volatilization caused by light source during the fluorescence microscopic imaging process, it is still a challenge to recode the in situ visualization of hierarchical self-assembly of the organic barcode microwires.’ was added on page 7, paragraph 2, lines 1-4 in the revised manuscript. The sentence ‘To further investigate the hierarchical self-assembly mechanism of the organic barcoded microwires, statistical fluorescence microscopy (FM) and SEM techniques’ is revised to be ‘To further investigate the hierarchical self-assembly mechanism of the organic barcoded microwires, the time-based statistical fluorescence microscopy (FM) and SEM techniques’ on page 7, paragraph 2, lines 4-7 in the revised manuscript.

Page 4, line 113: “These prepared BTP and BTBa in Figure S5.” This sentence is not complete. Further, use the word spectroscopy or spectra - do not mix both.

Response: Thanks for your valuable suggestion. The word spectroscopy or spectra should be not mixed both, which has been revised. Solid-state ^{13}C NMR results were reveals the chemical environment of the C atoms (*Angew. Chem. Int. Ed.* **2018**, *130*, 4027). According to solid-state ^{13}C NMR results (Figure R4a and R4b), after cocrystallization of BTB or BTP cocrystal, the chemical shift of BGP shows an obvious upfield shift from 125 to 127 or 128 ppm, indicating the increase of electron density (*Angew. Chem. Int. Ed.* **2020**, *60*, 6344). Meanwhile, the chemical shifts of TCNB at around 134 ppm and TFP at around 117 ppm exhibit downfield shifts. These observations demonstrate the π -charge-transfer from TCNB or TFP to BGP molecules, implying the existence of intermolecular CT interactions (*Adv. Mater.* **2016**, *28*, 5954). The electron spin resonance (ESR) spectrum of BTP exhibits a strong resonance signal with g factor of 2.0037 (Figure R4c). This result suggests the existence of unpaired electrons in BTP, which arises from the CT process (*ACS Nano* **2022**, *16*, 15000). Furthermore, the stronger ESR signal of BTB compared to that of BTP suggests a stronger CT interaction between BGP and TCNB than that between BGP and TFP. As depicted in Figure R4d and R4e, the characteristic peak of $\text{C}\equiv\text{N}$ stretching vibrations at 2244 cm^{-1} in TCNB and 2248 cm^{-1} in TFP is blue-shifted to 2242 cm^{-1} in BTB cocrystals and 2241 cm^{-1} in BTP cocrystals, respectively, suggesting a π -charge-transfer from BGP to TCNB or TFP (*Adv. Mater.* **2022**, *43*, 2107169). Compared to the constituent molecules, the BTP and BTB cocrystals exhibits a broader red-shifted peak at 500 and 620 nm as verified in Figure R4f, which is corresponding to the π -charge-

transfer from BGP to TFP or TCNB (*Angew. Chem. Int. Ed.* **2017**, *56*, 10352). As a short conclusion, the π -charge-transfer from BGP to TCNB or TFP in these prepared BTP and BTB cocrystals was verified by Fourier Transform Infrared (FT-IR) spectra, Solid-state ^{13}C NMR results, electron spin resonance (ESR) spectra, and diffuse reflection absorption spectra in Figure R4.

Figure R4. (a) and (b) Solid-state ^{13}C NMR results, (c) ESR spectra, (d and e) IR spectra, and (f) diffuse reflection absorption spectra of these organic BTP and BTB cocrystals.

Revisions: The sentence ‘These prepared BTP and BTB cocrystals presented the intermolecular CT transition from BGP to TCNB or TFP as verified by Fourier Transform Infrared (FT-IR) spectra, Solid-state ^{13}C NMR spectroscopy, electron spin resonance (ESR) spectra, and diffuse reflection absorption spectra in Figure S5.’ is revised to be ‘The π -charge-transfer from BGP to TCNB or TFP in these prepared BTP and BTB cocrystals was verified by Fourier Transform Infrared (FT-IR) spectra, Solid-state ^{13}C NMR results, electron spin resonance (ESR) spectra, and diffuse reflection absorption spectra in Figure S5.’ on page 4, paragraph 2, lines 15-18 in the revised manuscript. The corresponding exposition was added in the Supporting Information toward Figure S5.

Fig 3. g,h,l,n,- The ratios of BTB and BTP are not mentioned.

Response: Thanks for your valuable suggestion. The FM images of Figure R5f and R5g (Figure 3f and 3g) are corresponding to the same samples excited with the UV band (330-380 nm) and green band (500-550 nm), respectively. The corresponding ratio is 8:7. After transforming the excitation from UV to green light, strongly red-emissive core microwires coated by a weak red-emissive shell layer with a joint-like heterojunction were found as verified in Figure R5f and R5g (Figure 3f and 3g). The SEM image of Figure 3h is the organic barcoded microwires prepared by the same ratio of 8:7 with that in Figure R5f and R5g (Figure 3f and 3g). Similarly, the FM images of Figure R5l and R5n (Figure 3l and 3n) are corresponding the same samples in Figure

R5k and R5m (Figure 3k and 3m), respectively, which are excited with the green band. The corresponding ratios of 8:8 and 8:10 were added in Figure R5l and R5n.

Figure R5. (a) Schematic illustrations of the morphology evolution from barcoded to core/shell heterostructures via increasing the molar ratios between BTB and BTP cocrystals. (b-e) FM images of organic barcoded microwires excited with the UV band based on the different molar ratios between BTB and BTP: (b) 8:3, (c) 8:4, (d) 8:5 and (e) 8:6, scale bars are all 50 μm . (f, g) FM and (h) SEM images of organic core/shell microwires with an obvious heterojunction between the two shell layers at the center part. The scale bars of (f, g) and (h) are 50 and 20 μm , respectively. (i) FM and (j) SEM images of the typical heterojunction between the two shell layers at the center part with a scale bar of 2 μm . (k-n) FM images of organic core/shell microwires based on different molar ratios between BTB and BTP: (k, l) 8:8 and (m, n) 8:9. (f, k, and m) and (g, i, l, and n) were excited with the UV and green bands, respectively.

Revisions: The Figure 4 was replaced by Figure R5 in revised manuscript.

REVIEWERS' COMMENTS

Reviewer #3 (Remarks to the Author):

The authors corrected errors in the Figures and improved the manuscript.